# In-situ revitalizing end-of-life MBR membranes via a curtain-type dynamic membrane process

Yuxiang Liang[1], Yanqing Zhang[2], Fangfang Ye[1], Huan Feng[3], Pingli Li[1], Dongqing Zhan[1], Chenxuan Lou[1], Yangcheng Ding[1], Hai Xiang[1], Xiang Zhang[4], Baojing Gu [5], Fang Liu[6] ✉, Guosheng Shi [7] ✉, Fengchang Wu [8] ✉ & Huajun Feng [1] ✉

Membrane bioreactors (MBR), an industrial mainstay with over 15,000 installations worldwide, face severe sustainability challenges from frequent membrane replacement, generating over 500,000 tons of waste annually. Dynamic membranes, a promising alternative, are impeded in large-scale implementation by inherent issues of conventional flat-sheet designs, such as uneven flow distribution and performance instability. Herein, we propose a strategy that directly upcycles end-of-life hollow-fiber MBR membranes as substrates for curtain-type dynamic membranes (CTDMs). The curtain-type substrate layer rapidly forms a biofilm and attains stable effluent quality (turbidity <5 NTU) within 10 minutes. Exposing only 5% of the internal substrate surface for CTDM fabrication restores performance comparable to that of new MBR membranes while meeting discharge standards. Life-cycle assessment reveals a 1,070-fold reduction in carbon emissions (0.0025 vs. 2.67 kg $CO_2$ $m^{-2}$) and 99.9% lower environmental costs (0.0072 vs. 7.81 USD $m^{-2}$) compared to traditional membrane replacement. Scaling up this approach could mitigate 441 kt of $CO_2$ in China alone by 2035. This approach concurrently addresses two global crises, accumulating plastic waste from spent membranes and escalating energy costs of water treatment.

Membrane filtration, leveraging its unrivaled physical sieving precision, stands as a cornerstone of modern water purification, achieving near-complete (>99.99%) removal of suspended solids, pathogens, and macromolecular contaminants[1,2]. Among these technologies, membrane bioreactors (MBR) have emerged as an industrial mainstay through decades of material and engineering innovations, with global deployments exceeding 15,000 plants[3]. Compared to conventional activated sludge processes, MBRs deliver superior effluent quality (<1 NTU turbidity), compact footprint, and reduced sludge yield[4–6], fulfilling stringent discharge regulations with exceptional reliability.

[1]College of Environment and Resources, College of Carbon Neutral, Zhejiang A&F University, Hangzhou, China. [2]School of Environmental Science and Engineering, Zhejiang Gongshang University, Hangzhou, China. [3]Hangzhou Chaoteng Energy Technology Co. Ltd, Hangzhou, China. [4]College of Energy and Power Engineering, Xihua University, Chengdu, China. [5]College of Environmental and Resource Sciences, Zhejiang University, Hangzhou, China. [6]Zhejiang Province Key Think Tank: Institute of Ecological Civilization and Institute of Carbon Neutrality, Zhejiang A&F University, Hangzhou, China. [7]Shanghai Key Laboratory of Atomic Control and Application of Inorganic 2D Supermaterials, State Key Laboratory of Materials for Advanced Nuclear Energy, Shanghai Applied Radiation Institute, Shanghai University, Shanghai, China. [8]State Key Laboratory of Environmental Criteria and Risk Assessment, Chinese Research Academy of Environmental Sciences, Beijing, China. ✉e-mail: liufang@zafu.edu.cn; gsshi@shu.edu.cn; wufengchang@vip.skleg.cn; fenghuajun@zafu.edu.cn

However, two intrinsic limitations impede further MBR scalability. First, the pursuit of ultrahigh filtration standards leads to over-purification, a systemic energy burden that is disproportionate to actual water quality demands. Second, membranes suffer irreversible performance decay within complex activated sludge matrices, typically failing within 5 years due to pore fouling and mechanical abrasion caused by particulate debris[7,8]. Despite advances in fouling mitigation, ranging from pneumatic scouring to enzymatic treatments and electrokinetic controls[9–14], prolonged physical damage remains unavoidable. Notably, when the main supporting structure of MBR modules is intact, they are abandoned since only its functional layer is partly damaged, resulting in substantial membrane waste. The end-of-life (EoL) modules necessitate costly and environmentally hazardous replacement cycles, generating >500,000 tons/year of non-degradable membrane waste globally[7,15,16]. Effective resolutions to extend the service life of EoL MBR membranes remain lacking.

Dynamic membranes (DMs) technology serves as a promising alternative to conventional MBR technology. DMs employ low-cost, large-pore (10-50 µm) substrates to form in-situ cake layers that reject contaminants through biologically active filtration. Conventional DMs utilize woven metal or nonwoven materials in a flat-sheet configuration. However, heterogeneous flow distribution during surface area scaling compromises cake layer uniformity and stability[17,18]. Additionally, their low packing density ($50–100 \, \text{m}^2 \, \text{m}^{-3}$) necessitates large footprints[19]. The curtain-type membrane architecture fundamentally resolves scaling challenges through its fiber-array expansion strategy—achieving system scale-up by increasing fiber count rather than enlarging planar dimensions. However, critical knowledge gaps persist regarding their viability as dynamic membrane substrates.

In this work, we developed curtain-type dynamic membrane (CTDM) technology and evaluated its hydrodynamic advantages and filtration performance, with the aim of addressing the scalability challenges of flat-sheet membranes. A crucial observation is that over 90% of EoL MBR modules utilize hollow-fiber membranes, whose porous supports (with a pore size of ~20 µm) are structurally analogous to DMs substrates. However, current recycling strategies overlook this potential for functional repurposing. The modular, high-density ($>500 \, \text{m}^2 \, \text{m}^{-3}$) curtain-type design of CTDM inherently ensures uniform hydrodynamic conditions during scale-up through fiber multiplication. This stands in stark contrast to the flow-channeling effects that hinder area-expanded flat-sheet systems.

Based on these understandings, we propose a strategy that directly upcycles EoL hollow-fiber MBR membranes as substrates for CTDM, and performed comprehensive techno-economic and life-cycle analyses (LCA) to quantify carbon footprint reduction, resource recovery potential, and cost savings[20]. Notably, the CTDM technology successfully overcomes the technical bottlenecks of traditional flat-sheet dynamic membranes in large-scale applications. Building on the technology's underlying principles, this approach represents a strategy for regenerating EoL MBR membranes. This approach simultaneously addresses two global crises: the accumulating plastic waste from spent membranes and the escalating energy costs associated with water treatment, by converting waste into functional filtration resources.

## Results
### Hydrodynamic advantages and filtration performance of CTDMs
The structural differences between CTDMs and flat-sheet DMs fundamentally determine their scaling strategies and engineering applicability. Existing computational fluid dynamics (CFD) studies have extensively investigated the hydrodynamic advantages of flat-sheet and curtain-type membranes[21]. For flat-sheet membranes, their regular surface morphology and layout facilitate the extensive application of CFD simulations to optimize parameters such as wall shear stress and liquid flow velocity, thereby enhancing membrane antifouling performance[22,23]. For curtain-type membranes, they are well

recognized for their high packing density, which confers a compact footprint advantage, and CFD simulations are often employed to investigate the impacts of bubble scouring and mixed-liquor rheology on vortex-induced vibration[24]. However, for dynamic membranes, uniform transmembrane flow is more favorable for the development of a homogeneous biofilm. Therefore, this section seeks to perform a systematic comparative analysis of the flow velocity distribution of curtain-type and flat-sheet membranes during the scale-up process. As shown in Fig. 1a, the curtain-type membrane features a high-density fiber array structure, while Fig. 1b depicts the linear scaling characteristics of such curtain-type membranes. Although a single membrane fiber (1 m in length, with a surface area of $0.005 \, \text{m}^2$) exhibits a less uniform flow field compared to a laboratory-scale flat-sheet membrane of the same area owing to its considerable length, the curtain-type membrane achieves scale-up by increasing the number of membrane fibers (from 1 to 200), whereby each fiber retains an independent flow field. In contrast, the flat-sheet membrane relies on geometric scaling (Fig. 1c), which expands the lateral and longitudinal flow channels by over an order of magnitude, leading to uneven velocity distribution.

A comparative analysis of the axial flow velocities (Supplementary Fig. 1) shows that the curtain-type membrane sustains stable axial velocity fluctuations along the axial direction, while the flat-sheet membrane exhibits severe velocity decay, while velocity dropping abruptly within 10% of the distance to the outlet (a decrease of >94%). The comparative coefficient of variation (CV) of incremental flow velocity (Fig. 1d and Supplementary Fig. 2) effectively characterizes the flow velocity fluctuations within the two membrane configurations. When the membrane area is small ($0.005 \, \text{m}^2$), the CV of flat-sheet and curtain-type membranes is approximately 60–80%. However, as the flat-sheet membrane area scales up, its CV increases to over 250%, whereas that of curtain-type membranes remains unchanged. This result indicates the extreme non-uniformity of internal flow velocity distribution during the scale-up process of flat-sheet membranes. Collectively, the curtain-type membranes exhibit superior scalability advantages.

Beyond hydrodynamic advantages, CTDM filtration performance was also systematically evaluated. The filtration layer of CTDM utilizes curtain-type hollow fiber non-woven fabric as the substrate layer. The morphological evolution of the membrane surface during the filtration process was directly observed (Fig. 2b). After 1 h of filtration through the substrate layer, biofilm formation was detected within the substrate pores. During this process, the transmembrane pressure difference (TMP) was continuously maintained below 0.002 MPa, while the effluent turbidity decreased gradually, dropping to 5.5 NTU at 10 min and stabilizing at 3–4 NTU after 20 min (Fig. 2c, d). The particle size of suspended solids in the effluent also decreased from the initial 400 nm to 90 nm (Fig. 2e). These findings indicate that CTDM filtration adheres to the fundamental principles of conventional dynamic membrane filtration[18]. As activated sludge-containing mixed liquor passes through the porous substrate layer, suspended solids are deposited in the pores or on the surface, forming a secondary filter cake layer (Fig. 2a), which enables efficient solid-liquid separation by effectively filtering the influent. These results validate the feasibility of CTDM technology.

As the filtration time prolonged, a thick block-like biofilm formed on the surface, completely obscuring the substrate's structural characteristics. The increase in biofilm thickness from 0 to 7 h was mainly attributed to the accumulation of activated sludge (Fig. 2f). However, the rate of sludge accumulation gradually slowed down in the later stage, whereas the content of extracellular polymeric substances (EPS) increased rapidly, especially the data of bound extracellular polymeric substances (BEPS) (Fig. 2g). These results indicate that although the sludge cake plays a key role in sludge-water separation, excessive EPS secretion may cause fouling and increased transmembrane pressure during long-term operation[25]. Classic filtration models were employed

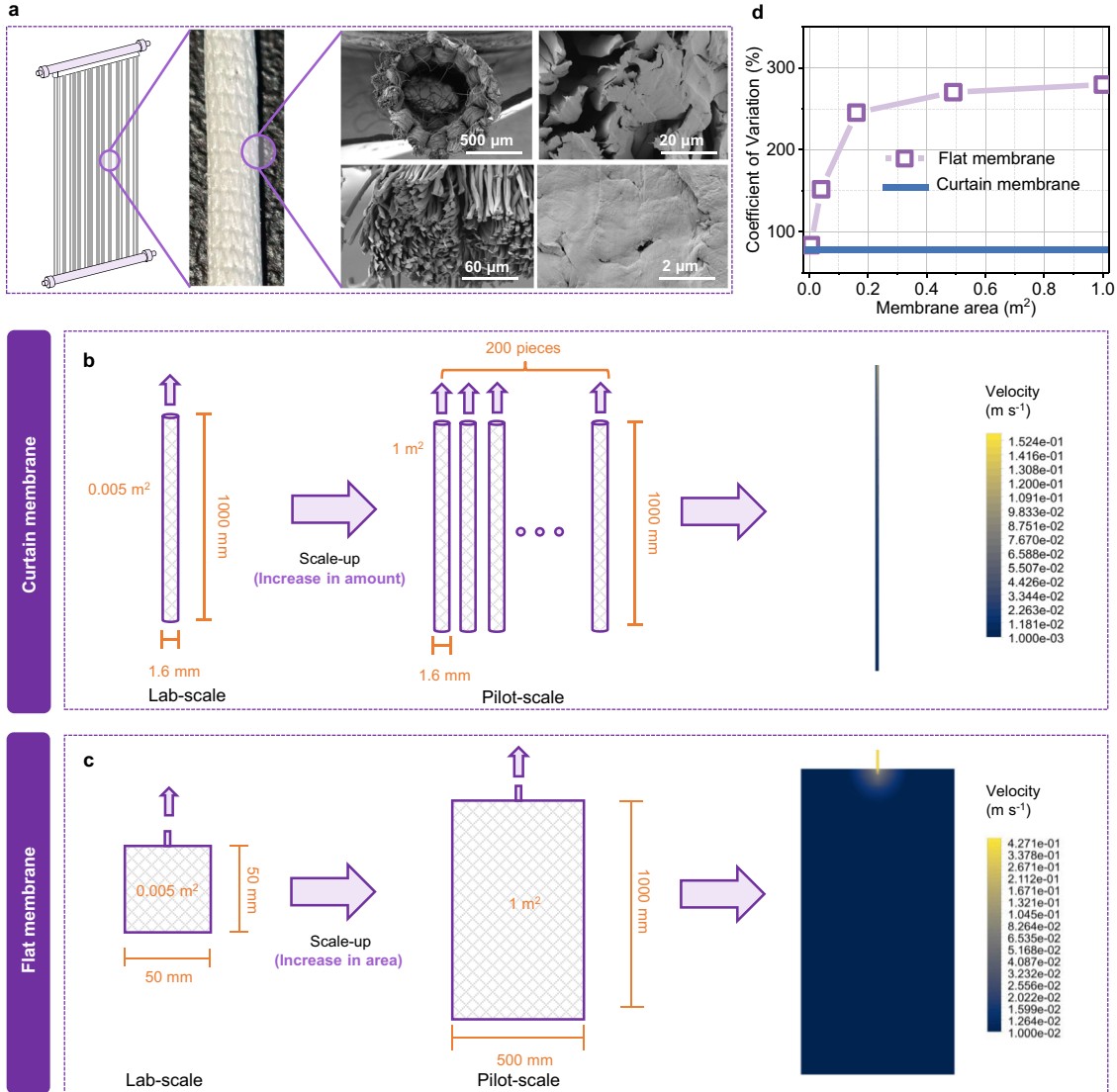

**Fig. 1 | Structural and hydrodynamic characteristics of curtain-type and flat dynamic membranes during scale-up. a** Structural features of the curtain-type dynamic membrane. **b**, **c** Flow velocity distributions in 1 m² scale modules of curtain-type and flat dynamic membranes. **d** Coefficients of variation for incremental flow velocity of curtain-type and flat dynamic membranes (data in Supplementary Fig. 2).

to analyze the flux decline curves (Fig. 2h). The analysis showed that the complete blocking model ($R^2 = 0.85$) was the optimal descriptor, which contradicts the conventional theory that cake filtration is dominant[26]. This discrepancy may arise from the fact that CTDM filtration involves the intrusion of activated sludge into membrane pores, whereas traditional flat-DMs rely on surface cake-layer formation. Consequently, CTDM fouling control demands pore-scale antifouling strategies—distinct from the shear scouring effective for surface cakes, thus offering foundational guidance for the development of CTDM cleaning protocols.

**Internal substrate layer reusable as CTDM substrates**
This study investigated the EoL membranes in MBR systems. Taking polyvinylidene fluoride (PVDF) membranes as an example, severe fouling was observed in membranes after 6 years of service (Supplementary Fig. 3), with the TMP increasing to 0.04 MPa within only 0.35 h, whereas new membranes could sustain stable operation for 5 h before reaching the critical pressure threshold (Fig. 3a). Furthermore, a significant reduction in surface functional groups was observed (e.g., carboxyl and amino groups), which were crucial for maintaining hydrophilicity and antifouling properties (Fig. 3b)[27,28]. In particular, the

intensity of the characteristic peak at 876 cm⁻¹, attributed to the out-of-plane rocking vibration of the C−F bond in the α-crystalline form of the PVDF molecular chain[29], was markedly reduced on the surface of the six-year-old membranes. This finding suggests that the key crystalline structure of the membrane had undergone irreversible transformation. Scanning electron microscopy (SEM) further confirmed that the filtration layer had completely lost its original porous structure after 6 years of use (Fig. 3c).

Notably, PVDF membranes are composed of an internal substrate layer and an external PVDF filtration layer (Supplementary Fig. 4)[30]. The PVDF filtration layer of EoL membranes exhibited extremely weak interfacial adhesion, allowing complete removal through manual peeling (finger pressure), adhesive tape stripping, or high-pressure water jet impact (Fig. 3d, e). These remaining internal substrate layers present an unexpected engineering opportunity, as they exhibit structural and functional equivalence to substrates used in CTDMs (Supplementary Fig. 5), thus providing an untapped resource-recycling pathway that converts membrane waste into circular economy value.

Building on this foundation, we systematically evaluated the feasibility of utilizing exposed substrate layers as CTDM substrates. The internal substrate layers feature a micron-scale structure and have

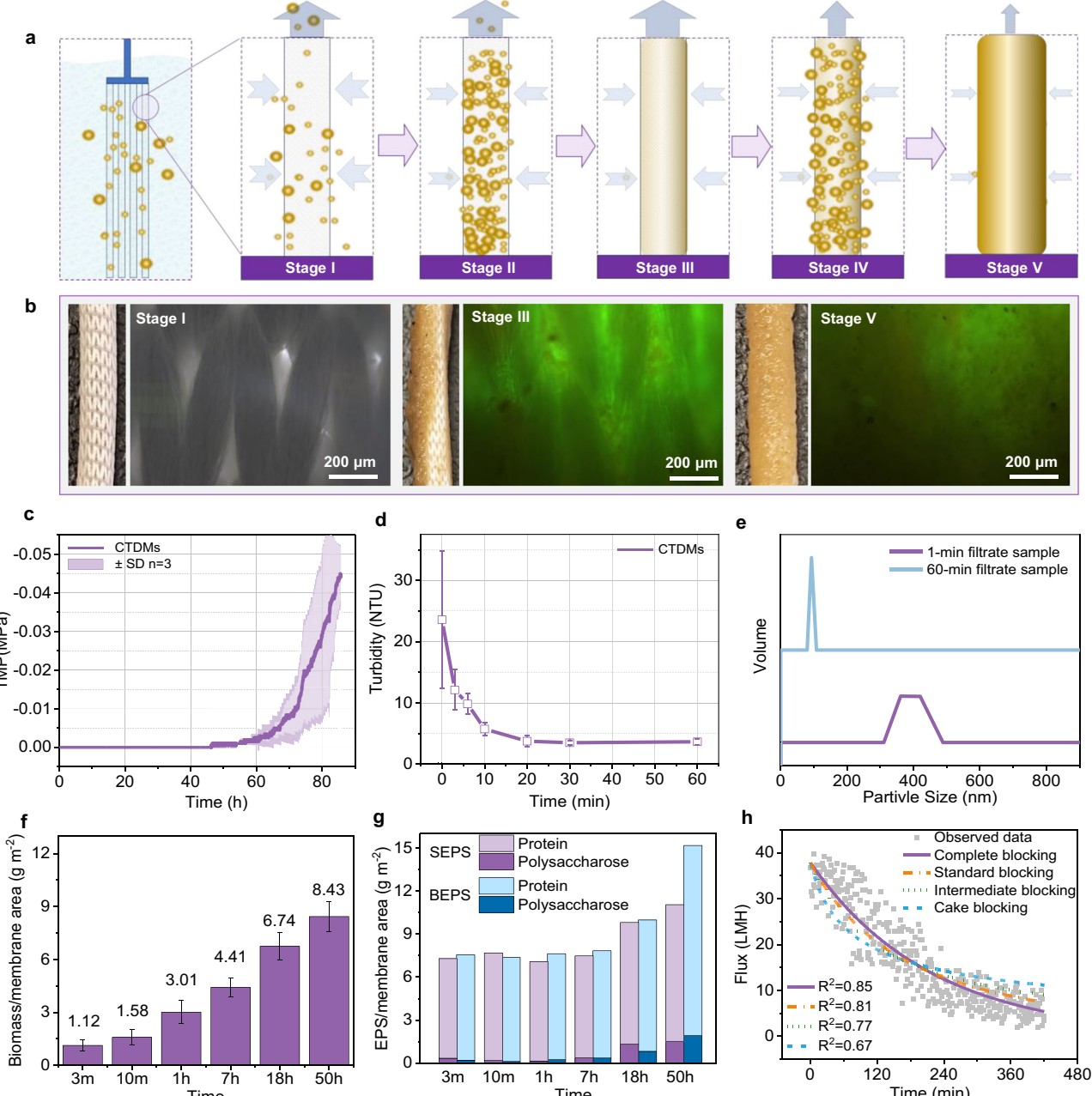

**Fig. 2 | Filtration mechanism and performance of curtain-type dynamic membranes (CTDMs). a** Schematic illustration of the filtration mechanism of CTDMs at different operational stages. **b** Photograph and live-dead fluorescence microscopy image of the membrane filament surface; live cells are stained green, and dead cells are stained red. **c, d** Trans-membrane pressure and effluent turbidity of the CTDM layer during operation in an MBR system at a flux of 15 LMH. **e** Effluent particle size distribution of the CTDM at different operating times. **f, g** Microbial content and extracellular polymeric substances (EPS) content on the membrane surface, respectively. **h** Filtration resistance model of the CTDM.

significantly lower porosity, surface area, and adsorption capacity. They lack special pollutant-intercepting functional groups, thereby lacking the ability to filter or adsorb microorganisms on their own (Supplementary Fig. 6). Prior to evaluation, the EoL membranes were pretreated with citric acid and sodium hypochlorite to remove surface impurities. Fourier transform infrared spectroscopy (FTIR) confirmed the absence of peaks of residual contaminants (e.g., EPS or metal complexes), while SEM observations verified a clean surface of the substrate layer, eliminating interference from residual pollutants for subsequent application.

Notably, a photograph of the cake layer formed on the exposed substrate layer of a discarded membrane (Fig. 3f) clearly shows that the

cake layer forms on the substrate surface. By contrast, no filter cake formed on the surface of the external PVDF filtration layer. This study employed Derjaguin Landau Verwey Overbeek (xDLVO) theory to elucidate the interfacial interactions between membrane components and activated sludge (Supplementary Fig. 7). Three cases were analyzed: Case A, EoL membrane versus activated sludge; Case B, substrate layer versus activated sludge; and Case C, filter cake layer versus activated sludge. Free energy profiles as a function of distance were calculated for three interaction types: Lewis acid-base (AB), van der Waals (LW), and electrostatic (EL) interactions. According to established interpretations, positive values indicate repulsive forces, while negative values indicate attraction forces[31].

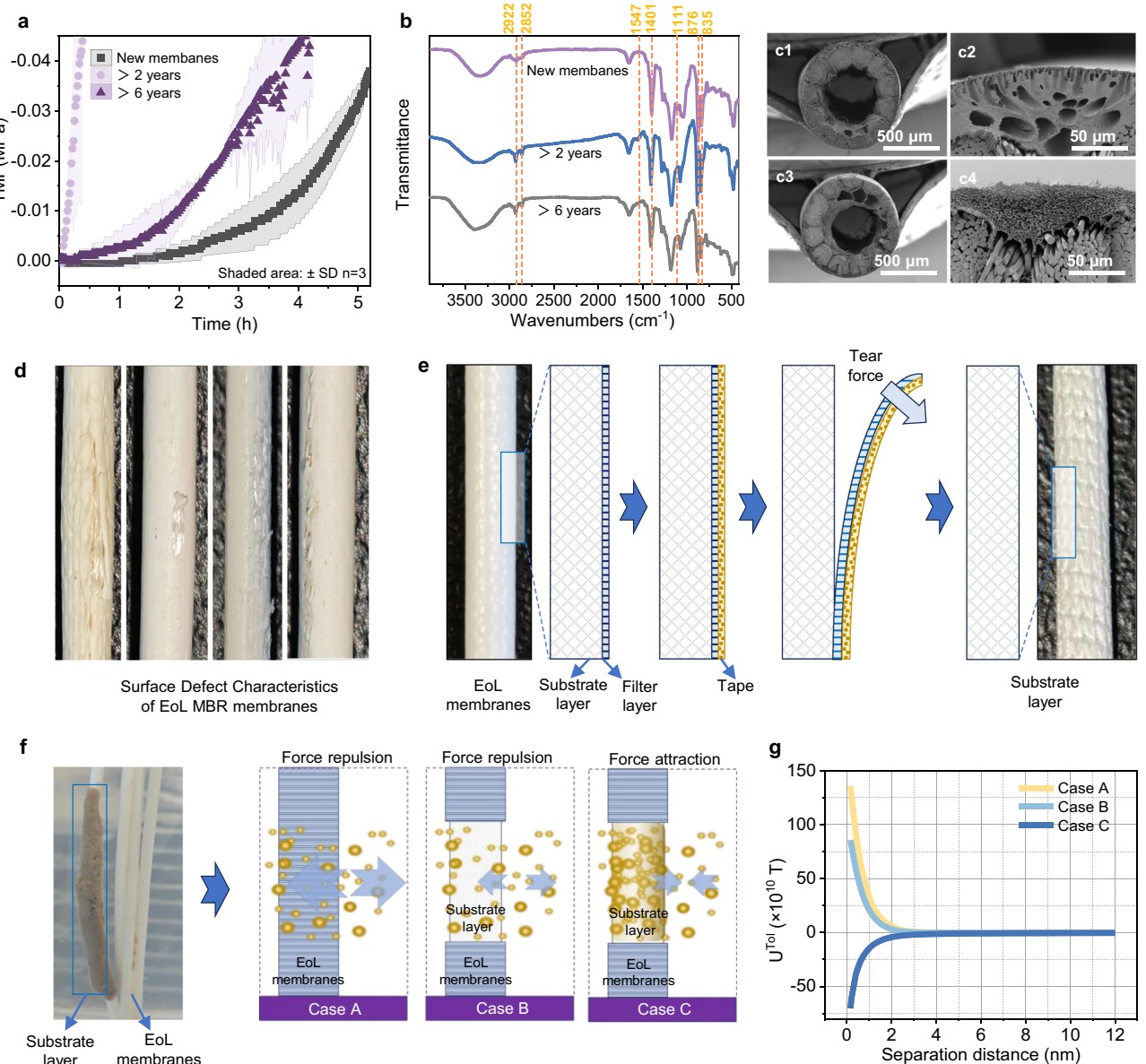

**Fig. 3 | Fabrication of CTDM substrates from end-of-life (EoL) MBR membranes.** **a** Trans-membrane pressure profiles of new and EoL membranes during operation in an MBR system at a flux of 15 LMH. **b** Surface functional group differences between new and EoL membranes. (**c1**–**c4**) Surface morphologies of new and EoL membranes; **c1**, **c2**: new membrane at different magnifications; **c3**, **c4**: EoL membrane at different magnifications. **d** Photograph of waste EoL membrane filaments with obvious surface damage and exposed inner support layer. **e** Schematic of the filtration layer being stripped from the EoL membrane surface using adhesive tape. **f** Intermolecular interactions under three different cases. Case A: interaction between EoL membrane and activated sludge; Case B: interaction between PVDF membrane substrate layer and activated sludge; Case C: interaction between bio-film and activated sludge. **g** Thermodynamic mechanism analysis of membrane surface fouling based on the xDLVO theory for the three cases above.

The results revealed that the surfaces of the EoL PVDF membranes exhibited relatively strong repulsive forces against activated sludge (Fig. 3g). In contrast, the exposed substrate layers demonstrated weaker repulsive interactions, making them more susceptible to sludge adsorption. Once the exposed supporting layers adsorb some microorganisms, their interaction with activated sludge becomes predominantly attractive due to enhanced LW and AB interactions. This promotes cake layer formation, which enables efficient solid-liquid separation by effectively filtering the influent. Overall, these findings indicate that the substrate layers have a higher affinity for activated sludge, facilitating the rapid formation of a secondary filtration layer. The results demonstrate that the exposed substrate layers of EoL membranes can effectively restores MBR flux through CTDM mechanisms.

## Flux recovery in MBRs using EoL membrane-based CTDMs

Controlled detachment of PVDF layers (1–50% exposure) demonstrates that exposed substrate layers recover hydraulic performance. However, at merely 1% exposure, TMP development mirrors that of new membranes (5 h to reach 0.04 MPa), with pressure escalation rate inversely proportional to exposed area (Fig. 4a). Crucially, initial effluent turbidity spikes (>5 NTU within 5 min) self-recover to <3 NTU within 5 min—outperforming the 30-min naturally settled supernatant (Fig. 4b). Figure 4c also illustrates that large-particle sludge is only discharged in the initial effluent stage. After 1 h of continuous operation, the effluent contains almost no large particle sludge, and the particle size is comparable to that of the intact PVDF membranes. The effluent turbidity is significantly correlated with the exposed area (Fig. 4d). Figure 4g illustrates the temporal evolution of membrane

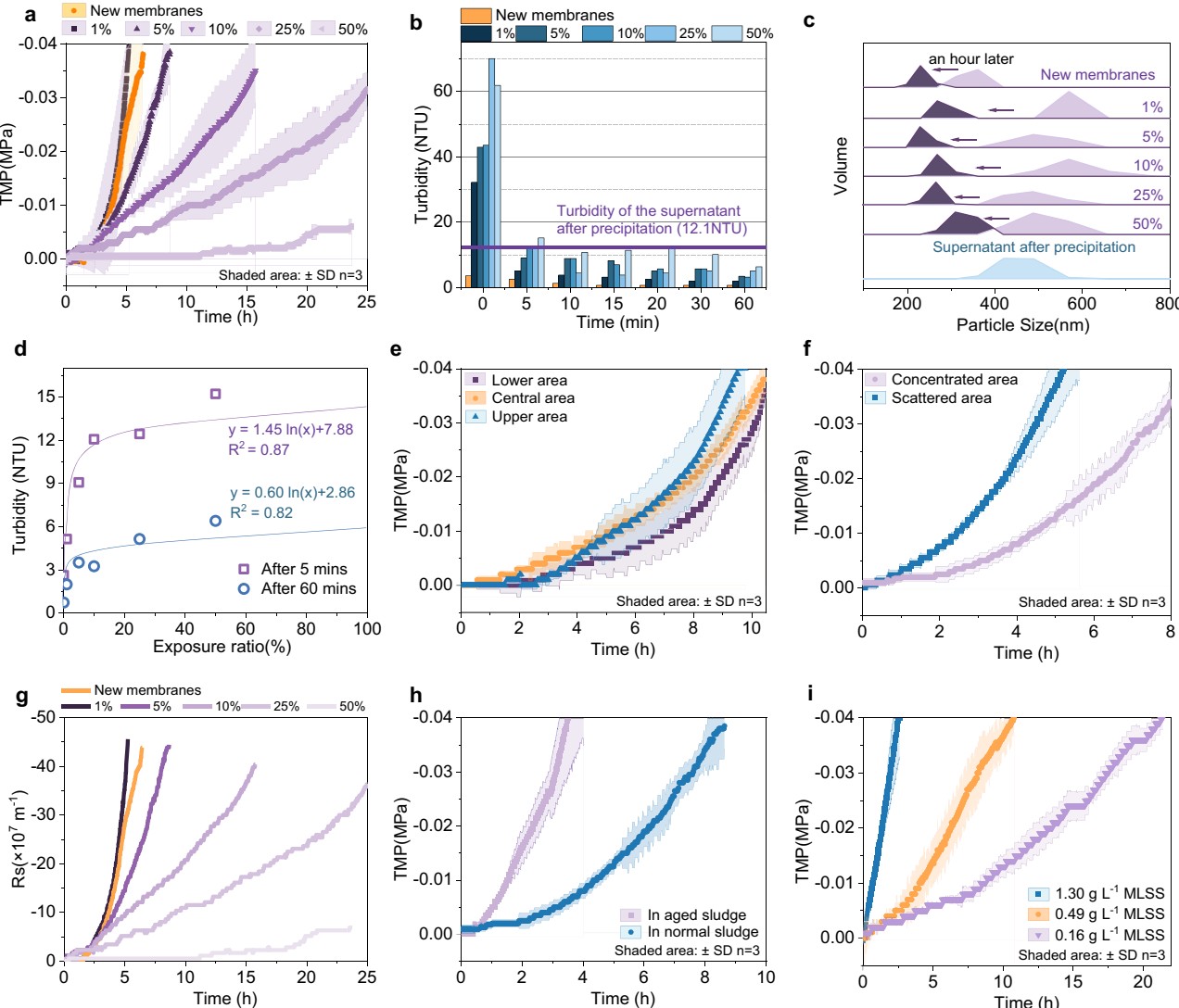

**Fig. 4 | Effects of the exposed membrane substrate layer on filtration performance. a** Trans-membrane pressure (TMP) profiles over time under different exposed substrate layer areas. **b** Effluent turbidity data of waste membranes with different exposed substrate layer areas, measured at staged sampling times. **c** Effluent particle size distributions of waste membranes with different exposed substrate layer areas at the initial stage and after 1 h of operation. **d** Correlation analysis between effluent turbidity and exposed substrate layer area at 5 min and 60 min of operation. **e, f** TMP evolution curves for membranes with 5% exposed substrate layer area, at different exposed positions and with different exposure dispersity. **g** Filtration resistance profiles over time under different exposed substrate layer areas. **h, i** TMP variations during filtration for membranes with 5% exposed substrate layer area, in activated sludge with different settling properties and different sludge concentrations, respectively.

filtration resistance (Rs) under varying exposed substrate areas. Notably, the growth rate of Rs decelerates significantly with increasing exposed area, indicating that the substrate layer possesses lower filtration resistance than the discarded PVDF layer. This fundamental difference in resistance directly accounts for the delayed TMP rise observed in Fig. 4a, as a lower overall resistance system maintains a more moderate TMP elevation.

These results strongly demonstrate that the substrate layer of EoL curtain membranes can serve as the substrate layer for DMs without compromising effluent turbidity compliance with discharge standards.

Furthermore, this study explored the effects of the exposed area and the activated sludge properties on filtration. The results show that the exposed position exerts little significant influence on the filtration flux (Fig. 4e), whereas more dispersed damage pattern is more conducive to extending the membrane fouling cycle (Fig. 4f). This phenomenon can be attributed to the fact that biofilms developing on smaller exposed substrate layer surfaces exhibit reduced thickness and adhesion, making them more susceptible to mechanical

detachment[32]. Consequently, such biofilms are less prone to evolving into an intact and robust filter cake layer through continued growth and consolidation. In addition, the substrate layer tends to be blocked in environments with poor sludge properties or high sludge concentration (Fig. 4h, i), because a higher concentration of EPS and sludge is more likely to block the substrate layer[12,17]. Collectively, the 5% exposure CTDM configuration sustains stable operation across typical MBR sludge conditions, validating EoL membrane upcycling as a robust strategy for flux recovery in MBR.

Surveys show that MBR systems are mainly used in municipal, industrial, agricultural wastewater treatment applications. To verify the technology's universality, we collected seven types of EoL membranes covering these scenarios for tests (Supplementary Tables 1-3). Results (Supplementary Figs. 8–13) indicate that exposing 5% of the internal substrate layer significantly extends the service life of all EoL membranes. Although their effluent turbidity is higher than that of intact MBR systems, it still meets discharge standards. Minor variations in improvement efficiency across membranes from different sources

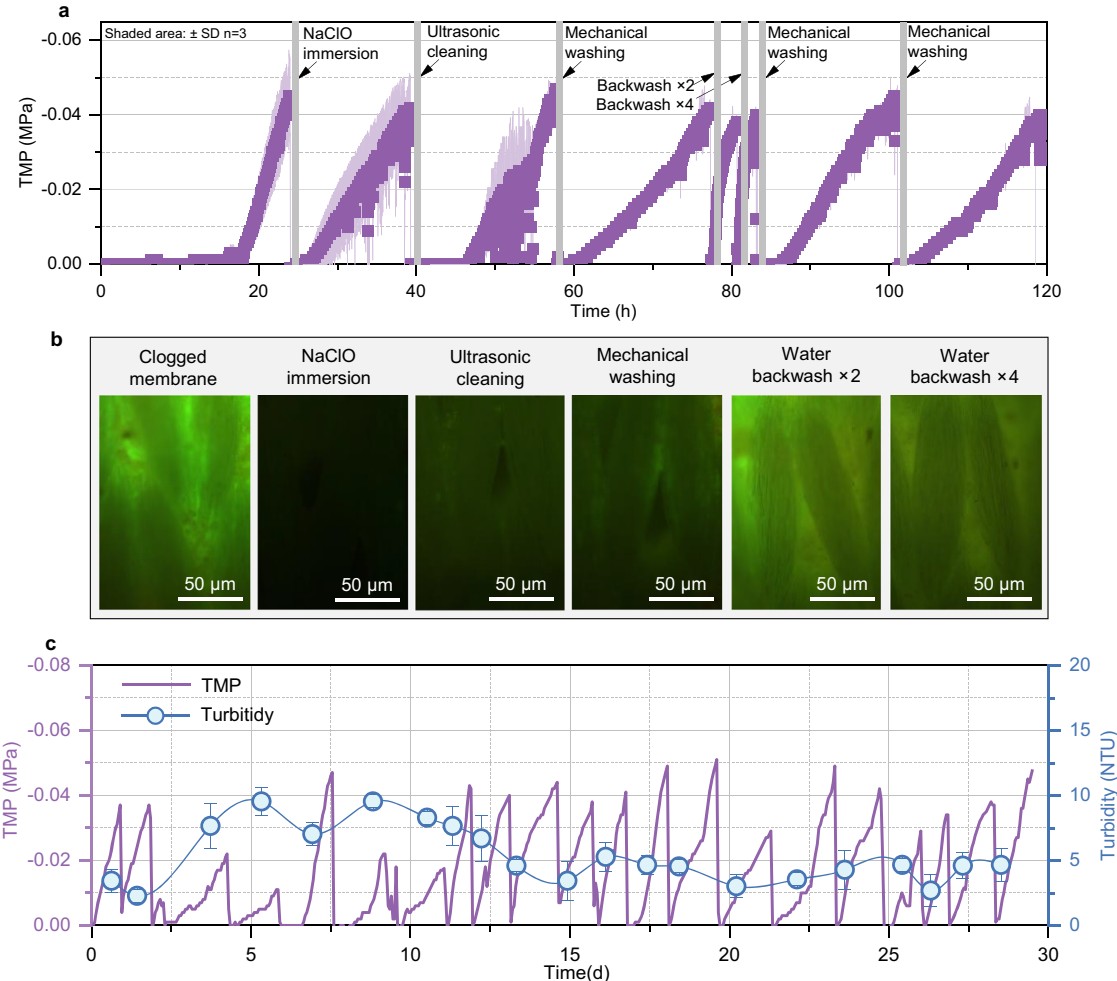

**Fig. 5 | Operational performance of CTDMs with 5% exposed substrate layer under the conventional MBR operation process. a** Effects of different backwash modes on the operational performance of CTDMs with a 5% membrane surface exposed area. **b** Fluorescence microscopy images showing microbial residues on the membrane surface after treatment with different cleaning methods. **c** Changes in TMP and effluent turbidity of CTDMs with a 5% membrane surface exposed area during 30 days of continuous domestic sewage treatment.

stem from differences in the extent of initial damage (e.g., PVDF layer peeling, pore blockage). These findings confirm the technology's general applicability in typical MBR scenarios, while also suggesting targeted performance tests are needed for specific industries to optimize the internal substrate layer exposure ratio. SEM observations reveal that all EoL membranes share similar morphological features, including internal substrate layer structure and PVDF peeling mode. Notably, EoL membranes from printing and dyeing, as well as pharmaceutical wastewater treatment, have more severe surface corrosion −likely causing natural internal substrate layer exposure during long-term service. TMP variations further internal substrate this: industrial EoL membranes have longer operation cycles than municipal and rural sewage membranes. This indirectly confirms that dynamic membranes were formed on the naturally exposed internal substrate layers of industrial EoL membranes during their original service. This finding has not been previously reported.

### Reliable long-term performance of CTDMs

The exposed substrate layer of membrane filaments effectively extends the membrane module's fouling cycle through a dynamic membrane formation mechanism. However, severe clogging, characterized by a transmembrane pressure difference exceeding 0.04 MPa, still occurs over time. Therefore, regular cleaning of the substrate layer is essential to maintain membrane functionality. In this study, various cleaning strategies commonly used in the MBR processes were evaluated to determine their effectiveness in restoring the flux of the membrane filament substrate layer. Methods including ultrasonic cleaning[33,34], mechanical scouring[35], and sodium hypochlorite soaking[36] effectively recovered membrane flux (Fig. 5a), and they reduced effluent turbidity to below 5 NTU within 30 minutes after cleaning (Supplementary Figs. 14 and 15). In contrast, backwashing with clear water at flow rates two and four times the normal level was ineffective in restoring membrane performance. After sodium hypochlorite soaking followed by cleaning, no microbial residues were observed on the substrate surface (Fig. 5b). Mechanical scouring and ultrasonic cleaning successfully removed the cake layer, although some residual sludge remained on the substrate surface. Backwashing, however, only dislodged the larger, blocky components of the filter cake and failed to remove microorganisms embedded within or adhered to the substrate layer, rendering the membrane more prone to rapid re-fouling. These findings aligned with the pore-blocking model (Fig. 2h), demonstrating that CTDMs achieve filtration not merely through surface cake-layer formation but via pervasive biofilm development within membrane pores. Owing to this deep-seated fouling mechanism, backwashing proves ineffective for CTDM maintenance, while alternative strategies including mechanical scouring and chemical cleaning remain viable for sustained operation.

To assess the operational stability of EoL membranes, a 30-day continuous filtration test was conducted using actual domestic wastewater (Fig. 5c). The PVDF membrane with 5% surface exposure maintained its filtration capacity throughout the test, despite undergoing repeated cleaning. From Days 1 to 10, the membrane was cleaned with sodium hypochlorite soaking, while from Days 11–30, mechanical cleaning was employed. During this period, the effluent turbidity remained consistently within acceptable discharge limits for municipal wastewater. A distinct pattern emerged: sodium hypochlorite treatment resulted in gradual TMP recovery with higher effluent turbidity (5–10 NTU), whereas mechanical cleaning produced faster TMP recovery and significantly lower turbidity (<5 NTU). This suggests that excessive chemical cleaning may inhibit dynamic membrane formation, while mechanical methods more effectively internal substrate filtration performance.

To validate the practical feasibility of the CTDM technology in real-world municipal wastewater scenarios, a 45-day pilot-scale experiment was conducted at the Phase IV Facility of Yuhang Wastewater Treatment Plant (Hangzhou, Zhejiang, China), using a 20-ton-per-day Anaerobic–Anoxic–Oxic–MBR treatment unit (Supplementary Fig. 16). The CTDM system was assembled from 6-year-old waste MBR membranes with 5% internal substrate layer exposure, and its performance was compared with a conventional intact MBR system. Results showed that while CTDM effluent turbidity was slightly higher than that of intact MBR membranes (still meeting general application requirements), there was no significant difference in COD, $NH_4^+$-N, TP, and SS removal—attributed to pollutant removal primarily relying on microbial metabolism in the bioreactor (Supplementary Fig. 16c). Furthermore, Dynamic membranes have been demonstrated to effectively remove various types of pollutants, including sulfonamide antibiotics, heavy metals, dissolved organic matter, and sub-micron-sized polystyrene microplastics, in wastewater treatment processes[37–39].

Notably, long-term operation verified two key stability aspects: the core woven skeleton of the internal substrate layer remained structurally intact (no skeleton damage, pore collapse, or structural deformation. Supplementary Fig. 17), and the 5% peeled PVDF interface (separation between the PVDF layer and internal substrate layer) showed no unintended detachment or expansion (Supplementary Fig. 18). These results confirm that under practical operational conditions, the CTDM system maintains both internal substrate layer structural stability and PVDF-internal substrate layer interface stability, while sustaining consistent pollutant removal and filtration performance. In addition, after dynamic membrane formation, similarly high effluent quality has been reported for supports with pore sizes ranging from 5 to 50 μm[40]. This supports the finding that an effective dynamic layer can still form on the internal substrate layer despite potential minor changes in its pore size or morphology.

As the CTDM technology relies on the existing MBR operational framework, it must be compatible with conventional MBR cleaning regimes (intermittent backwashing, low-frequency chemical cleaning, and mild mechanical scrubbing). To evaluate whether conventional MBR cleaning methods affect CTDM's stability, experiments were conducted targeting the internal substrate layer structure and PVDF-internal substrate layer interface: a 48-hour cyclic ultrasonic test (10 min on/10 min off) and a 48-hour NaClO immersion test. Results showed neither the substrate layer structure nor the peeled PVDF interface exhibited unintended changes (e.g., interface detachment, substrate layer deformation; Supplementary Figs. 19, 20), indicating that conventional MBR cleaning methods have relatively minor impacts on both aspects of CTDM stability. Although our experiments demonstrate that conventional MBR cleaning methods (low-frequency chemical cleaning and mechanical scrubbing) have a relatively minor impact on PVDF layer integrity under the tested conditions, potential cumulative effects may arise from long-term, high-frequency, or high-intensity cleaning in practical applications. Additionally, while continuous PVDF peeling might occur during extended operation, results from Fig. 4a–d suggest that even with greater peeling, the CTDM system's filtration performance remains stable, further supporting its practical applicability.

The issue of transient turbidity exceeding regulatory standards during the initial operational phase can be effectively addressed through engineering interventions. Two technically feasible strategies during system startup include: (1) temporarily containing non-compliant effluent, such as diverting it to controlled discharge systems or repurposing it for non-critical applications, and (2) alternating backwashing cycles among multiple membrane units to achieve dilution-based turbidity reduction. Currently, substrate layers used in dynamic membrane systems are typically made from polypropylene (PP) and polyethylene terephthalate (PET). According to existing reports, these materials have demonstrated a service life of approximately five years in wastewater treatment applications[7,41]. Although their chemical resistance is lower than that of PVDF, PP and PET substrate layers offer greater thickness and superior mechanical properties, including higher tensile and compressive strength[40]. In conclusion, substrate layers provide sustained and reliable filtration performance in dynamic MBR systems.

## Mitigation of environmental impacts

Based on the above findings, we propose a strategy for reusing EoL membranes (Fig. 6a, Option II). When the membrane module in the MBR system is damaged, we suggest continuing to use the damaged EoL membrane by artificially exposing part of the membrane surface, thereby increasing its filtration flux via a CTDM process. Although the final effluent turbidity in Option II is marginally higher (3–4 NTU) compared with that in the MBR system (<1 NTU), it still meets the requirements of various international discharge standards. The traditional method is to replace the membrane module with a new one to restore the filtration performance of the MBR process (Fig. 6a, Option I). The disassembled EoL membranes are generally disposed of as solid waste through incineration or landfilling. The production of new membranes requires a large amount of chemicals such as PVDF and Dimethylacetamide (Fig. 6b). Membrane production is energy- and labor-intensive, while also generating substantial production wastewater and solid waste, which poses significant threats to both economic benefits and the environment (Supplementary Fig. 21).

Taking China as a case study, the MBR process was mainly promoted and adopted on a large scale starting from 2011 (Fig. 7a), with its capacity rapidly increased from 2.30 million tons per day to 38 million tons per day in 2024, making China one of the countries with the most extensive use of the MBR process globally[5]. According to the prediction of Oracle Crystal Ball (based on the double exponential smoothing method), the application scale of the MBR process will be further expanded to 60 million tons per day by 2030. At present, the membrane modules for the MBR process in China are mainly independently produced by domestic enterprises. As the most important membrane material, the production scale of PVDF membranes has also increased rapidly since 2011 and is expected to reach 100 million $m^2$ by 2030 (Fig. 7c). Commonly used membrane modules in membrane bioreactors (MBRs) mainly include curtain-type membranes, flat-sheet membranes and other structural forms. Among these, curtain-type membranes are the most widely used mainstream membrane type in current MBR systems, accounting for 78.9% of the market share (Supplementary Table 1). For curtain-type membrane products, the introduction of substrate layers can significantly enhance the mechanical strength of the membrane structure and extend the service life of membrane modules. This technical advantage has made it a mainstream design scheme—among existing commercial curtain-type membrane products, those with substrate layers account for more than 81.9% (Supplementary Table 2). Based on the above data, it can be calculated that the proportion of curtain-type membranes with

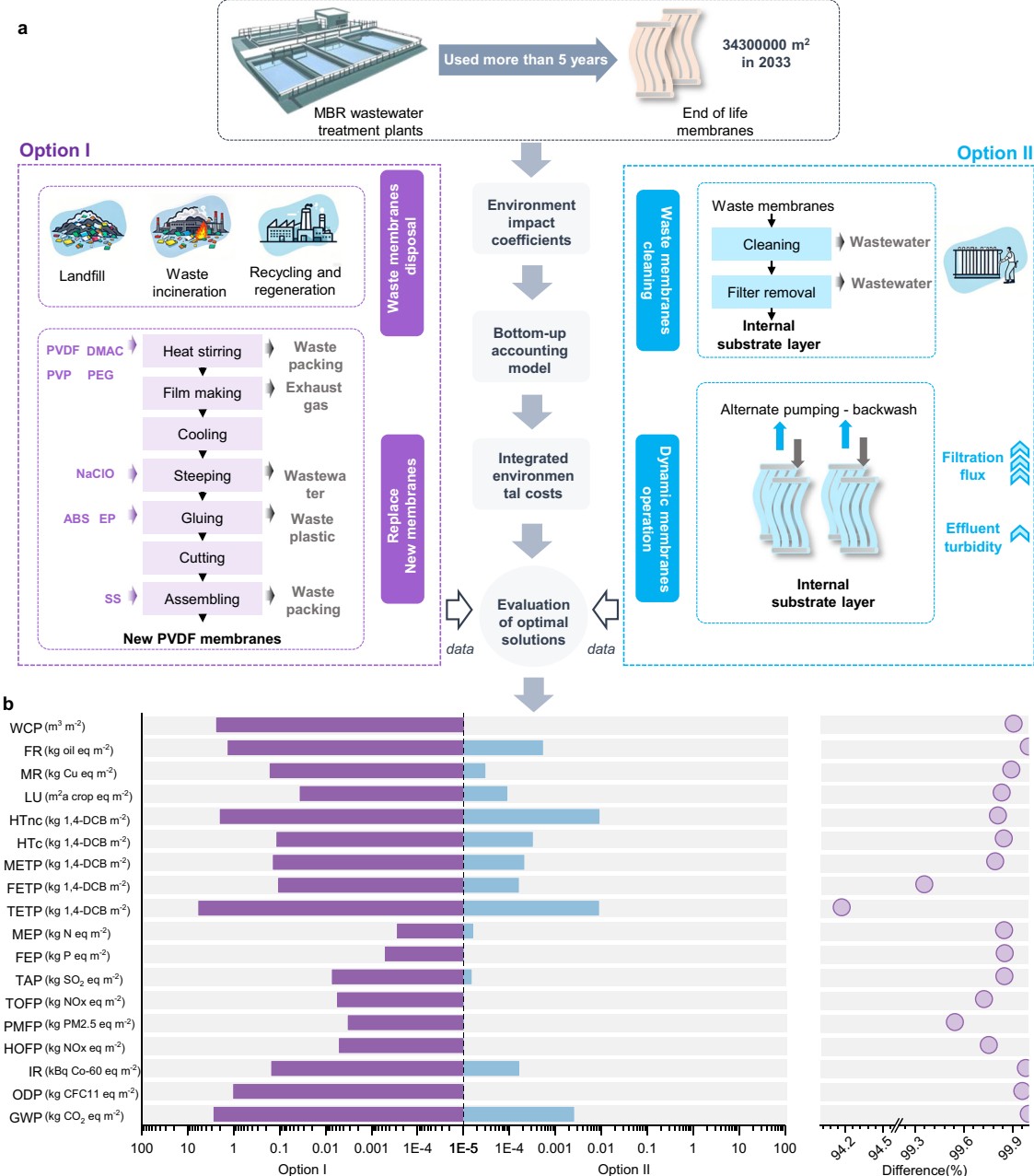

**Fig. 6 | Life cycle assessment (LCA) of traditional and proposed end-of-life (EoL) MBR membrane treatment strategies. a** Schematic of two EoL membrane treatment pathways: Option I (conventional method) – EoL membranes disposed via landfill or incineration, with replacement by new MBR membranes; Option II (proposed method in this study)–cleaned EoL membranes reused after stripping 5% of the surface filtration layer to expose the substrate layer for an additional operational lifecycle. **b** Comparative LCA results between Option I and Option II, evaluated using seventeen characteristic indicators related to the potential environmental impacts of the technical processes.

substrate layers in the total production capacity of MBR membrane modules is approximately 64.6%. According to this calculation, the projected output of curtain-type MBR membranes with substrate layers in China will reach 64.6 million m² by 2030.

Since industrial wastewater is often mixed into municipal wastewater in China, resulting in relatively high concentrations of pollutants such as oxidants, acidic and alkaline substances[42]. Under the action of strong oxidants, the molecular chains of PVDF membranes are prone to cleavage, leading to damage to the membrane pore structure[43]. In addition, numerous small-scale MBR wastewater treatment plants in urban and rural areas of China[44]. Due to the lack of professional membrane module maintenance personnel, the service life of membrane modules is significantly shortened (often less than 5 years).

Therefore, the membrane replacement ratio in China is extremely high (Fig. 7b). Based on an average service life of 5 years, it is estimated that the discarded membrane modules will reach 15 million m² by 2025, and the area of discarded membrane modules is expected to reach 29 million m² by 2035. The cost of replacing membrane modules is substantial, with the replacement cost ranging from 0.022 to 0.058 USD per ton (Fig. 7d).

In this study, seventeen characteristic indicators associated with the potential environmental impacts of the technical process were chosen for evaluation. These indicators mainly cover three aspects: carbon emissions, pollutant emissions, and toxic health impacts, namely Global warming (GWP), Stratospheric ozone depletion (ODP), Ionizing radiation (IR), Ozone formation in the troposphere (to

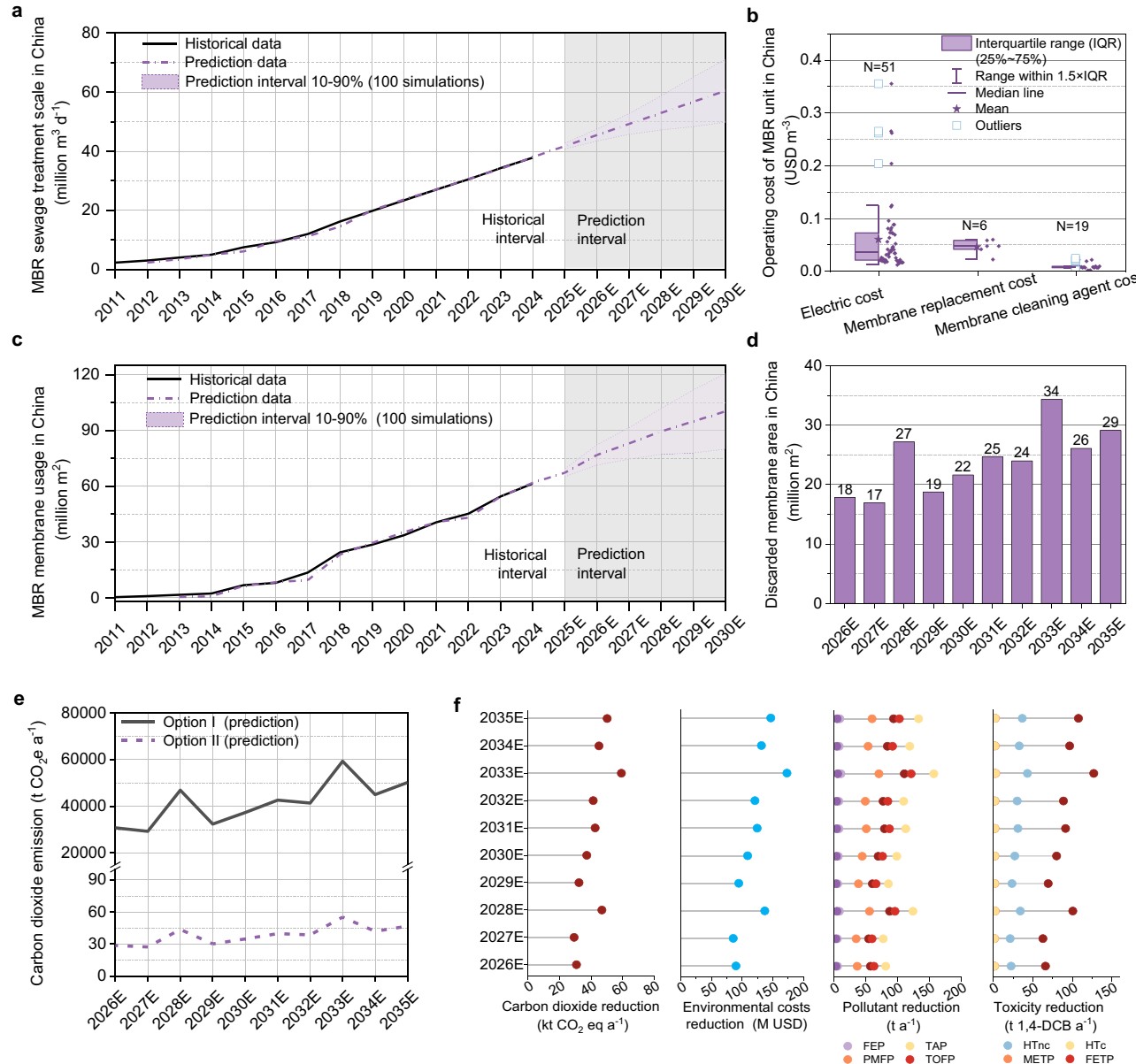

**Fig. 7 | The EoL membranes in MBR Systems and the reduction potential under different treatment scenarios. a** Statistical analysis of the overall scale of the MBR process during 2011-2024 and prediction of the overall scale of the MBR process during 2024–2030 in China. **b** Investigation into the composition of MBR process operation costs. **c** Statistical analysis of the overall scale of membrane production during 2011-2024 and prediction of the overall scale of membrane production during 2024–2030 in China. **d** Prediction of the quantity of EoL membranes during 2026-2035. **e** Predictions of carbon dioxide emissions of Option II and Option I during 2026–2035. **f** Predictions of reduction in carbon dioxide emissions, environmental costs, toxicity levels and pollutant emissions by taking Option II instead of Option I.

distinguish the repeated term), Human health (HOFP), Fine particulate matter formation (PMFP), Terrestrial ecosystems (TOFP), Terrestrial acidification (TAP), Freshwater eutrophication (FEP), Marine eutrophication (MEP), Terrestrial ecotoxicity (TETP), Freshwater ecotoxicity (FETP), Marine ecotoxicity (METP), Human carcinogenic toxicity (HTc), Human non-carcinogenic toxicity (HTnc), Land use (LU), Mineral resource scarcity (MR), Fossil resource scarcity (FR), and Water consumption (WCP).

In terms of carbon emissions, each square meter of membrane in Option II results in the emission of 0.0025 kg of $CO_2$, while each square meter of the updated membrane in Option I needs to emit 2.67 kg of $CO_2$, which is 1070 times the amount of $CO_2$ emitted by Option II. Even when considering the existing technology that proposes to recycle the PVDF material on the surface of the EoL

membranes through melting and extraction to prepare new PVDF membranes[45], its carbon emissions are still much higher than those of Option II. Based on this, the study further predicts the environmental benefits in terms of carbon emissions obtained by replacing the existing Option I with Option II in the next 10 years (Fig. 7e). Implementing this replacement will reduce will reduce 441 k-tons of $CO_2$ in China from 2026 to 2035.

In Option I, the production of PVDF membranes is energy-intensive and generates significant amounts of wastes (Supplementary Tables 4, 5). Option II repurposes EoL membranes instead of using traditional landfilling or incineration treatment methods, preventing the environmental pollution caused by landfilling or incineration. Simultaneously, it also reduces the demand for new membrane materials. The monetary results revealed a dramatic reduction in

environmental costs between membrane technologies (Supplementary Fig. 22 and Tables. 6–9). Option II achieved a monetary cost of 0.0072 USD per square meter, two orders of magnitude lower than that of Option I (7.81 USD m$^{-2}$). Economic projections (Fig. 7f) indicate that replacing Option I with Option II will yield cumulative cost savings of 1.29 billion USD over the period from 2026 to 2035, representing a transformative reduction in both operational and environmental expenditures.

In addition, Option II can reduce hundreds of tons of pollutants (mainly TAP, PMFP, HOFP, TOFP) each year (Fig. 7f). More importantly, Option I also poses substantial toxic risks to the environment and human health. A large number of chemicals used in the membrane production process are suspected carcinogenic. At temperatures exceeding the melting point, polymer fragments are released, and the decomposition products may include aldehydes, alcohols, organic acids, hydrogen chloride, and trace amounts of hydrocarbons[46]. Furthermore, pollutants such as nitrogen oxides and particulate matter are emitted during the combustion of fossil fuels and waste. Therefore, Option I has a greater impact on TETP and HTnc. It is projected that replacing the existing Option I with Option II can comprehensively reduce 150 k-tons of 1,4-DCB eq by 2035. In summary, Option II is a highly promising carbon-reduction strategy that minimizes waste and pollution while delivering substantial economic and environmental advantages.

## Discussion

We have developed a curtain-type hollow fiber dynamic membrane technology, which can quickly form a filtration layer within 10 min, enabling stable effluent quality (turbidity <5 NTU). This technology utilizes the interception capability of hollow fiber membranes to form a cake layer via the entrapment of activated sludge on the surface, acting as a secondary filtration barrier that enables efficient retention of sludge particles. Although the effluent turbidity meets discharge standards, its moderate inferiority to conventional MBR systems limits its application to wastewater treatment plants, with a risk of non-compliance where higher water quality is required. In addition, this technology has successfully addressed the technical bottlenecks of traditional flat-sheet dynamic membranes in large-scale applications, such as uneven flow distribution and heterogeneous cake layer filtration during large-scale use.

Based on this, we propose a upcycling strategy for EoL MBR membranes involving the removal of the fragile separation layer from the surface of spent hollow-fiber membranes to expose the internal substrate for CTDM fabrication. However, this technology is restricted to curtain-type dynamic membranes with an internal substrate layer, as the structural integrity and stability of the substrate are prerequisites for forming a uniform secondary filtration layer. Notably, only 5% surface exposure can restore performance to a level comparable to that of brand-new MBR membranes, while meeting discharge standards. This approach simultaneously addresses two global crises, accumulating plastic waste from spent membranes and escalating energy costs associated with water treatment, by transforming waste into functional filtration assets.

Building upon these findings, this study proposes a sustainable reuse strategy for EoL membranes in MBR systems (Option II). The core value of Option II lies in its ability to extend the service life of membrane modules, thereby reducing the frequency and cost of replacements. This approach avoids the substantial material, energy, and labor costs associated with new membrane production, as well as the associated generation of solid and liquid waste. Option II also delivers significant environmental advantages, including a substantial reduction in carbon emissions and avoidance of pollutants and hazardous by-products typically generated during membrane manufacturing and disposal. Notably, Option II holds strong application potential, particularly in developing countries where economic

constraints and operational sustainability are key priorities. The ability to lower both capital and operational expenditures without compromising treatment performance aligns well with the strategic needs of resource-limited regions.

## Methods

### Construction and operation

The reactor was a plastic reactor with dimensions of $14.0 \times 9.0 \times 19.0$ cm and an effective volume of 1.5 L. An aeration diffuser was installed at the bottom of the reactor, and the air was supplied by an aeration pump (SOBO Air Pump, SB-988). The filtration layer of CTDM employed hollow fiber polypropylene non-woven fabric with 20–25 μm pore size (diameter: 1.6 mm, thickness: 0.4 mm) as the substrate layer (Hangzhou Tianchuang Environmental Technology Co., Ltd). The EoL PVDF membranes were sourced from the third phase of Yuhang Wastewater Treatment Plant (used for more than 6 years), Linping Water Purification Plant (used for more than 4 years), and the fourth phase of Yuhang Wastewater Treatment Plant (used for more than 2 years) in Hangzhou City, Zhejiang Province, China. Prior to use, the EoL membranes underwent sequential chemical pretreatment to remove surface and pore-blocking contaminants: first, they were soaked in a 2% citric acid (Analytical Grade, AR) solution at room temperature ($22 \pm 3$ °C) for 12 h to dissolve metal cation precipitates; subsequently, they were immersed in a 0.5% NaClO (effective Cl ≥20.0%) solution for an additional 12 h to degrade microbial extra-cellular polymeric substances (EPS) and organic pollutants. During each soaking step, the solution was gently agitated every 6 h to ensure uniform contact between the membrane and the cleaning agent. After chemical pretreatment, the membranes were thoroughly rinsed with deionized water until the effluent pH reached neutrality (6.8–7.2), then dried at 40 °C for 24 h prior to subsequent PVDF layer peeling and experimental assembly. To artificially simulate EoL membranes with controlled degrees of damage, the PVDF layer was peeled off using adhesive tape to expose fixed surface areas, with the exposed fractions set at 1%, 5%, 10%, 25%, and 50%, respectively.

Activated sludge was operated in a sequencing batch reactor (SBR) mode with a hydraulic retention time (HRT) of 6 h. The system was continuously aerated from the bottom to supply oxygen (air-to-water ratio = 8), which facilitated mixing of reactor contents and regulated sludge cake formation on the membrane surface. Two peristaltic pumps were configured: one was used for effluent discharge from the hollow-fiber dynamic membrane, and the other was used for influent feeding at the same flow rate to maintain a constant liquid level in the reactor. TMP was monitored using a pressure transmitter (Asmik MIK-P300) installed on the effluent pipeline to reflect membrane fouling status.

Membrane modules were considered fully fouled when the TMP reached −0.04 MPa, at which point they were disassembled, cleaned, and reinstalled for subsequent operation. Four cleaning methods were evaluated: ultrasonic cleaning, mechanical scouring, sodium hypochlorite soaking, and clear water backwashing. Ultrasonic cleaning involved removing the membrane module, placing it in an ultrasonic cleaner for 10 min of ultrasonic treatment, and then reassembling it for testing. Mechanical scouring involved removing the membrane module and removing accumulated sludge on the membrane surface with flowing water. Sodium hypochlorite soaking involved placing the membrane module in a 15% (v/v) sodium hypochlorite solution for 10 min. Clear water backwashing involved reversing the direction of the peristaltic pump and injecting clear water into the membrane module in the reverse direction. "×2" and "×4" denoted backwashing flow rates of 2 times and 4 times the effluent flow rate, respectively.

Supplementary Methods 1-3 detail the filtration performance tests of end-of-life (EoL) MBR membranes from various industrial sectors, the pilot-scale experimental setup and operation, and the ultrasonic and NaClO immersion stability tests, respectively.

## Experimental wastewater and sludge

The activated sludge was sourced from the aeration tank of Tangqi Wastewater Treatment Plant in in Hangzhou City, Zhejiang Province, China. The concentration of the added sludge was $3500 \pm 500$ mg L$^{-1}$, and the sludge SV30 was 20%. Simulated domestic sewage: Sucrose (AR) was used to prepare a COD concentration of 200 mg L$^{-1}$, NH$_4$Cl (AR) was used to prepare an ammonia nitrogen concentration of 40 mg L$^{-1}$, KH$_2$PO$_4$ (AR) was used to prepare a total phosphorus concentration of 5 mg L$^{-1}$, and trace elements (1 mL L$^{-1}$) were added, with a total volume of 1 L. The trace elements included H$_3$BO$_4$ (150 mg L$^{-1}$), ZnSO$_4$·7H$_2$O (120 mg L$^{-1}$), MnCl$_2$·7H$_2$O (120 mg L$^{-1}$), CuSO$_4$·5H$_2$O (30 mg L$^{-1}$), Na$_2$MoO$_4$ (65 mg L$^{-1}$), NiCl$_2$ (50 mg L$^{-1}$), CoCl$_2$·6H$_2$O (210 mg L$^{-1}$), and KI (30 mg L$^{-1}$). All trace element reagents were of AR. The actual wastewater was collected from the influent of the fourth phase of Yuhang Wastewater Treatment Plant in Hangzhou City, Zhejiang Province, China. The system operated at room temperature ($22 \pm 3$°C).

## Analytical Methods

The surface and cross-sectional morphologies of membrane materials were characterized by a scanning electron microscope (SEM, ZEISS Sigma 300, Germany) equipped with an Oxford Xplore 50 energy-dispersive X-ray spectrometer (EDS) for elemental composition analysis (C, O, F and other main elements). For sample preparation, the membrane samples were directly affixed to conductive adhesive tapes. A Quorum SC7620 sputtering coater was used for conductive gold coating with a pure gold target at a current of 10 mA for 45 s to eliminate charging effects. For morphological observation, the accelerating voltage was set to 3 kV with the SE2 secondary electron detector. For EDS elemental mapping and quantitative analysis, the accelerating voltage was adjusted to 15 kV at a fixed working distance of 8.5 mm.

The surface functional groups of membrane materials were identified by an FTIR (Thermo Fisher Scientific Nicolet iS20, USA) in attenuated total reflection (ATR) mode, and all tests were conducted in a dry environment to avoid moisture interference. The ATR accessory was first placed in the optical path of the spectrometer to scan the air background for baseline correction. Subsequently, the outer surface of the membrane sample was tightly attached to the crystal surface of the ATR accessory to ensure full contact. The test parameters were set as follows: spectral resolution of 4 cm$^{-1}$, cumulative scan times of 32, and spectral range of 400–4000 cm$^{-1}$. The infrared spectra were collected in transmittance mode for subsequent functional group analysis.

The specific surface area and pore structure of membrane materials were determined by an automatic specific surface area and porosity analyzer (Micromeritics ASAP 2460, USA) under the mesopore testing mode (for simultaneous analysis of specific surface area and pore size distribution). High-purity N$_2$ was used as the adsorption gas, and all tests were performed at 77 K (liquid nitrogen temperature). Prior to testing, the membrane samples were pretreated on the degassing station matched with the instrument: vacuum degassing was conducted at 120 °C for 6 h to remove adsorbed water and surface impurities, and the samples were confirmed to be structurally stable without decomposition at this degassing temperature. After pretreatment, the N$_2$ adsorption–desorption isotherms of the samples were collected by the instrument, and the total specific surface area of the membrane materials was calculated by the BET method.

The formation process of DM under a fixed flux was observed using a fluorescence microscope (Nexcope NIB-900, China). The cell dye used was the Cell Viability/Cytotoxicity Assay Kit (Beyotime Biotech Inc DMAO/PI C2030S, China). A small section of DM was cut and placed on a glass slide, and 1 μL of the staining working solution was added. It was incubated at 37 °C in the dark for 15 min. After that, it was placed under a fluorescence microscope to observe the staining effect.

After the membrane had been in operation for a period of time, the sludge cake on the membrane surface was removed by ultrasonic treatment. Microbial content was determined via the dry weight method in parallel with EPS extraction. The biofilm on the membrane surface was ultrasonically detached and collected into a pre-weighed weighing bottle, which was then dried at 105 °C for 2 h, cooled to room temperature in a desiccator, and weighed with an electronic analytical balance. For EPS, the sludge sample was centrifuged at 4 °C and $3820 \times g$ for 10 min using a refrigerated centrifuge (TGL-16 M, Hunan, China). Then, the supernatant was filtered through a 0.22 μm filter to obtain the soluble organic fraction as SEPS. The recovered residue was resuspended in a buffer solution to the original volume, and then it was used to extract BEPS by heating at 60 °C for 30 min and centrifuging at $10,610 \times g$ and 4 °C for 20 min. The supernatant was filtered through a 0.22 μm filter. The contents of proteins and polysaccharides in the EPS samples were analyzed. The protein content was determined by the BCA method using the BCA Protein Assay Kit (Beyotime Biotech P0010S, China). The polysaccharide content was tested by the phenol-sulfuric acid method. A 1.0-mL sample was taken, and 1.0 mL of distilled water was added. Then, 1.0 mL of 6% phenol was added, followed by the rapid addition of 5.0 mL of concentrated sulfuric acid. The mixture was shaken on a vortex mixer for thorough mixing, allowed to stand for 30 min, and the absorbance at 490 nm was measured. The measured absorbance was then substituted into the standard curve to determine the polysaccharide concentration.

An appropriate amount of the sludge mixture in the reactor was filtered by suction to form a filter cake layer of 1–3 mm. The contact angle was measured using a contact angle/surface tension tester (SDC 350KS, Kunshan Shengding, China). To calculate the various surface tension components of the solid surface, contact angle measurements were carried out using three typical probe liquids (ultrapure water, glycerol, and n-hexadecane) to determine the contact angles of these three probe liquids on the same sludge filter cake layer. The droplet size was 2 μL, each measurement lasted for 60 s, and the calculation was carried out using the stable value. The Zeta potential of the sludge mixture was determined using a solid surface potentiometer (Thermo Scientific K-Alpha, USA). The activated sludge mixture was directly sampled from the reactor without additional pretreatment to maintain its original physical and chemical properties. All measurements were performed at room temperature ($22 \pm 3$ °C).

The turbidity was measured using the standard method[47]. The particle size distribution of the effluent from DM was measured using a dynamic light scattering instrument (Anton Paar, Litesizer 500, Austria) with measuring dishes corresponding to the particle sizes, respectively, and the measurement range was 0–1000 μm.

## Statistical analysis

Numerical simulation of flow fields is based on CFD technology, which describes flow field information in the entire computational domain by iteratively solving the flow parameters of discrete points in the flow domain (Supplementary Tables 10, 11 and Supplementary Method 4). Fouling model equations for various operational modes are shown in Supplementary Table 12 and Supplementary Method 5. Relevant data were fitted using Origin software.

**xDLVO analysis.** The xDLVO theory extends the classical DLVO theory, adding the Lewis acid–base interaction (AB) to the van der Waals (LW) and electrostatic (EL) effects of the DLVO theory[30]. The specific methods are detailed in the supplementary materials (Supplementary Tables 13, 14 and Supplementary Method 6).

**Scale Prediction Analysis.** The data on the operation scale of the MBR process and the production scale of MBR membranes in China from 2011 to 2024 are from the consulting company (https://www.gonyn.com)[5,6], and the MBR operation data are sourced from literature[5,6] and the operation data of the sewage treatment plants under Yuhang Water Affairs in Hangzhou City, Zhejiang Province, China for the

period 2022–2024. For the period from 2025 to 2030, through the predictor function in the Oracle Crystal Ball (11.1.2.4) software, by analyzing the data distribution from 2011 to 2024, the optimal non-seasonal and non-accidental simulation method was selected, and performed 100 simulation runs (Supplementary Tables 15, 16). Questionnaire Survey on Market Share of Supported Curtain-Type MBR Membranes Among Major Manufacturers in China can be found in Supplementary Methods 7.

**Life cycle inventory.** The data on production-related resource consumption and pollutant emissions from membrane modules were obtained from the environmental impact assessment (EIA) report of a membrane production enterprise in Jiangsu Province. The resource consumption and pollution emissions required for membrane regeneration and repair in this study were estimated through experimental data (Supplementary Tables 4–7). The data were analyzed using OpenLCA software[48,49]. The monetization factors for seventeen characteristic indicators were estimated according to our previous research[21], considering that each of these environmental indicators is caused by multiple environmental pollutants (Supplementary Tables 8, 9). In China, the environmental taxes are defined for each pollutant, expressed as USD per pollutant equivalent (converted to euros in the final results after estimation).

### Statistics and reproducibility

Membrane samples used in this study, including those collected from full-scale wastewater treatment plants and membrane operation systems in other industrial sectors, were randomly selected and covered different service life cycles to ensure the representativeness of test samples and avoid selection bias. No statistical method was used to predetermine the sample size, and the experimental setting was in accordance with the established experimental norms in the fields of membrane separation and wastewater treatment research. All determinations of key performance indicators including transmembrane pressure, effluent turbidity, extracellular polymeric substances content and particle size distribution were conducted via three independent parallel experiments. All raw experimental data generated during the research were fully recorded and retained, with no data excluded from subsequent statistical analyses to maintain the integrity and representativeness of the research dataset.

The experimental procedures were not randomized in this study due to the fixed operational parameters of the membrane filtration system and the inherent structural characteristics of the curtain-type dynamic membrane module. Investigators were not blinded to allocation during experiments and outcome assessment given the clear identification of membrane sample sources and experimental treatment groups for real-time monitoring of variables. All raw experimental data were statistically processed and analyzed using Microsoft Excel, with data fitting and experimental figure plotting completed via Origin software. All error bars presented in the figures represent the standard deviation (SD) of three independent parallel replicates ($n = 3$). Error bars for datasets with a large number of data points are represented in shaded form. The consistency of results from three independent parallel experiments fully verified the reproducibility of the experimental methods in this study and the reliability of the corresponding research conclusions.

### Data availability

The data generated in this study are provided in the main text, Supplementary Information and Source Data file. The Source Data generated in this study have been deposited in the ScienceDB database under accession code https://doi.org/10.57760/sciencedb.27734.

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

## Acknowledgements

We acknowledge support from "Leading Goose" R&D Program of Zhejiang (2024C03111) to H.X., and the National Natural Science Foundation of China (12574227) to G.S. We acknowledge Professor Li Jun's team from Zhejiang University of Technology for providing the aerobic granular sludge.

## Author contributions

Y.L. and H.F. conceived the study and edited the manuscript. Y.Z., H.F., P.L., F.Y., D.Z., C.L., H.X., Y.D., and X.Z. assisted with data collection and the initial analysis. B.G., F.L., G.S. and F.W. supervised the analysis.

## Competing interests

The authors declare no competing interests.
