## [Transparent Peer Review file · Nature Communications]

In-situ Revitalizing End-of-Life MBR Membranes via a Curtain-Type Dynamic Membrane Process

Corresponding Author: Professor Huajun Feng

Version 0:

Reviewer comments:

Reviewer #1

(Remarks to the Author)

This is a study on a highly relevant topic for separation engineering with membranes. The intensification of secondary (biological) treatment of wastewater by membrane bioreactor (MBR) technology has become technically and commercially very successful, with a steady growth rate of number and size of installations world-wide. The authors claim and provide evidence that after the membranes in an MBR have reached their end-of-life (EoL), an alternative to replacement and disposal of the EoL membrane can be established, that would largely increase sustainability of the MBR technology. They show convincingly, that removing (part of) the polymeric ultrafiltration (UF) membrane from the macroporous support and continuing the filtration will lead to a dynamic membrane of biomass on the macroporous support that has similar (but not identical!) filter capability compared to the original UF membrane on that same support (cf. Fig. 3). Toward practical implementation, they show that by partial removal of the UF membrane layer from a fouled MBR membrane, the performance in terms of permeance and turbidity removal can be adjusted to meet the criteria for running the MBR and releasing the effluent (Figs. 4 and 5). However, effluent turbidity was higher than with an UF membrane. Based on the lab-scale feasibility, they project via LCA to potential sustainability improvement for the industry, assuming that all MBR systems would change from single use and replacement to continued use via curtain-type dynamic membrane process. Overall, I find the work well performed and relevant to in terms of showing new opportunities. However, there are aspects that need more differentiated consideration.

First of all, only part of the membranes used for MBR are capillary membranes; world-wide there is also a significant market share of flat-sheet membrane systems. Nevertheless, I agree that capillary membranes have the largest share. However, a significant share of capillary membranes that are used in MBR are unsupported membranes! Whether the more robust supported, but much thicker, or the unsupported thinner capillary membranes are used in a big MBR installations depends on the type of wastewater and the foot-print requirements. The method proposed here will (may) only work for supported membranes. Hence, the projection via LCA to reduction potential is too much simplified. I do not have numbers at hand, but I would guess that it is about 50:50 world-wide (the authors may dig deeper to justify their assumptions).

Second, the permeate quality will be significantly different for an intact UF membrane including a fouled but not damaged one (barrier pore size depending on membrane type 20 to 50 nm), compared to the biomass filter cake that is formed on an open macroporous support (here the authors talk about 20 μm). Authors only focus on turbidity removal, and this is overall less for the dynamic membrane compared to the UF membrane. Hence, if feasibility and cost of operation is not only judged based on quantity (flux, permeance), but also based on quality (of the permeate) the novel methodology may not be option in many cases.

And those are few more specific comments.

The title is not well chosen with: "revitalizing ... the vitality"

The language needs careful revision, throughout the paper. For example, grammar/wording of first sentence of Abstract is confuse. Also, the use of tense (past vs. present) is inconsistent. For example, obviously, facts that motivate the study should be presented in present tense: "The former (MBR) face sustainability challenges ..." instead of "The former (MBR) faced sustainability challenges ..."

p. 8: Reference 28 is used to imply that PVDF membranes (for MBR) consist of an internal substrate layer and an external PVDF layer. However, that paper by Sulaiman et al. is devoted to: "A sustainable and stable supported liquid membrane (SLM) extraction of nickel was developed via impregnation of sustainable liquid membrane in the composite membrane

support consisting of polyvinylidene fluoride (PVDF) and sulfonated poly (ether ether ketone) (SPEEK)". Except for the use of PVDF, that work has nothing to do with the state-of-the-art of capillary membranes for MBR application! This is another illustration of my general statement above, that the authors do apparently not care much about important details, regarding relevant membrane structures for concrete application scenarios.

p. 9: The input used for the estimations based on xDLVO (results in Fig. 3g) remains unclear.

Reviewer #2

(Remarks to the Author)

The manuscript entitled "In-situ Revitalizing Sustainable Vitality of End-of-Life MBR Membranes via a Curtain-Type Dynamic Membrane Process" presents a novel and environmentally relevant strategy to upcycle end-of-life hollow-fiber membranes from conventional membrane bioreactors (MBRs) into curtain-type dynamic membranes (CTDMs). The authors demonstrate that controlled detachment of the PVDF layer (at 5%) can restore membrane performance to levels comparable to new MBR membranes while maintaining scalability. Furthermore, the inclusion of a life-cycle assessment effectively supports the claimed environmental and economic benefits, emphasizing the potential of this approach to enhance membrane sustainability and reduce waste generation.

The subject is relevant and addresses a pressing challenge in the field of membrane technology and wastewater treatment. This study is particularly interesting because it proposes an innovative upcycling approach that not only extends membrane lifespan and reduces environmental impact but also overcomes common scalability issues through the development of curtain-type dynamic membranes. However, several important questions and concerns remain unresolved. Therefore, in my opinion, the current manuscript cannot be recommended for publication in Nature Communications until the authors have adequately addressed the following comments and remarks.

Concern#1: The information presented regarding the hydrodynamic advantages and filtration performance is not novel to the field of membrane technology, as the fundamental differences between hollow-fiber (curtain-type) and flat-sheet membranes have been extensively documented in the literature. The authors appear to reiterate these well-established concepts without providing sufficient citation support, giving the impression of contributing original knowledge, although the novelty of this contribution is not clearly demonstrated.

Concern#2: One of the main unresolved issues in this paper concerns the variability of the recycled membranes, which has not been evaluated. Discarded membranes originating from different water sources and type of usage may exhibit variations in both fouling and scaling characteristics, surface properties, or mechanical integrity, which could potentially affect both the ease of PVDF layer removal and the filtration performance at 5% (or other) substrate layer exposure. Such variability could therefore limit the straightforward implementation of the proposed upcycling strategy.

Concern#3: It is unclear whether, once the PVDF layer is partially removed to achieve the desired exposure, the exposed substrate remains stable over time. Hydraulic forces and biological activity could enlarge these gaps, potentially leading to excessive removal of the PVDF layer and resulting in unintended substrate exposure, such as the 50% exposure observed in Figure 4, thereby affecting membrane performance.

Concern#4: In addition, the choice of mechanical or chemical cleaning methods could significantly impact operational performance by affecting how much of the PVDF layer is removed. Different cleaning regimes may lead to uneven substrate exposure, which could alter hydraulic flow, biofilm development, and overall membrane filtration efficiency.

Concern#5: It is not clear how the authors intend to address the issue that EoL membranes may contain traces of chemical cleaning agents or tightly bound contaminants, which are difficult to remove completely. Such residues could interfere with biofilm formation or potentially compromise permeate quality, but the manuscript does not explain how these risks would be mitigated.

On the other hand, minor remarks should be also considered.

1. There is a typo in Figure 2e, see "partivle".

Version 1:

Reviewer comments:

Reviewer #1

(Remarks to the Author)

The authors have convincingly addressed the comments and question, by additional experiments and data analysis, and they have revised the manuscript including supporting information carefully. I recommend that the paper may now be published.

Reviewer #2

(Remarks to the Author)

Dear Editor,

I have carefully reviewed the revised manuscript and the authors' responses to my comments. All my concerns have been adequately addressed, and the manuscript has been significantly improved through clear explanations, additional experiments, and the inclusion of information derived from the survey. I therefore recommend the manuscript for publication.

Best regards,

Responses to reviewers

Manuscript ID: NCOMMS-25-54060

Title: In-situ Revitalizing End-of-Life MBR Membranes via a Curtain-Type Dynamic Membrane Process

Authors: Yuxiang Liang, Yanqing Zhang, Fangfang Ye, Huan Feng, Pingli Li, Dongqing Zhan, Chenxuan Lou, Yangcheng Ding, Hai Xiang, Xiang Zhang, Baojing Gu, Fang Liu, Guosheng Shi, Fengchang Wu, Huajun Feng

Reviewer 1:

Q1: This is a study on a highly relevant topic for separation engineering with membranes. The intensification of secondary (biological) treatment of wastewater by membrane bioreactor (MBR) technology has become technically and commercially very successful, with a steady growth rate of number and size of installations world-wide. The authors claim and provide evidence that after the membranes in an MBR have reached their end-of-life (EoL), an alternative to replacement and disposal of the EoL membrane can be established, that would largely increase sustainability of the MBR technology. They show convincingly, that removing (part of) the polymeric ultrafiltration (UF) membrane from the macroporous support and continuing the filtration will lead to a dynamic membrane of biomass on the macroporous support that has similar (but not identical!) filter capability compared to the original UF membrane on that same support (cf. Fig. 3). Toward practical implementation, they show that by partial removal of the UF membrane layer from a fouled MBR membrane, the performance in terms of permeance and turbidity removal can be adjusted to meet the criteria for running the MBR and releasing the effluent (Figs. 4 and 5). However, effluent turbidity was higher than with an UF membrane. Based on the lab-scale feasibility, they project via LCA to potential sustainability improvement for the industry, assuming that all MBR systems would change from single use and replacement to continued use via curtain-type dynamic membrane process. Overall, I find the work well performed and relevant to in terms of showing new opportunities. However, there are aspects that need more differentiated consideration.

Response: Thank you for your positive evaluation of our work and constructive feedback. We fully agree with your point that the permeate quality (e.g., turbidity) of the curtain-type dynamic membrane (CTDM) is slightly lower than that of intact ultrafiltration (UF) membranes, and that the LCA projection needs to be refined by considering membrane type heterogeneity. We have addressed these limitations by: (1) supplementing additional water quality indices (COD, ammonia nitrogen, total

phosphorus, and suspended solids) to verify that CTDM meets discharge standards; (2) revising the LCA model to reflect the market share of supported hollow-fiber membranes; and (3) clarifying the applicable scenarios of the proposed technology. These revisions enhance the practical relevance and scientific rigor of the study.

In addition, we sincerely apologize for the incorrect citations in the initial manuscript. To rectify this, we rechecked all references, verified their relevance, and replaced mismatched ones with literature aligned with our study focus.

All comments have been addressed through manuscript revisions, supplementary data additions, and careful corrections. Grammatical revisions are highlighted in green, while content modifications are highlighted in yellow.

Q2: First of all, only part of the membranes used for MBR are capillary membranes; world-wide there is also a significant market share of flat-sheet membrane systems. Nevertheless, I agree that capillary membranes have the largest share. However, a significant share of capillary membranes that are used in MBR are unsupported membranes! Whether the more robust supported, but much thicker, or the unsupported thinner capillary membranes are used in a big MBR installations depends on the type of wastewater and the foot-print requirements. The method proposed here will (may) only work for supported membranes. Hence, the projection via LCA to reduction potential is too much simplified. I do not have numbers at hand, but I would guess that it is about 50:50 world-wide (the authors may dig deeper to justify their assumptions).

Response: We sincerely apologize for the over-simplification in the initial LCA projection, and your comment is crucial for improving the accuracy of our environmental impact assessment. To address this issue, we have taken the following measures:

1. Supplemented market share data: We first consulted with relevant enterprises and professors in this field, and their consensus aligns with your inference that most MBR membranes are curtain-type membranes with support layers. We apologize for being unable to find relevant data in industry reports or literature. Thus, we resorted to a questionnaire survey to assess the proportion of curtain-type membranes with support layers. We distributed 15 questionnaires to major MBR membrane manufacturers in China and received 10 valid responses. These selected enterprises account for over 30% of China's total MBR membrane production scale, a proportion that sufficiently underpins the rationality and validity of the survey data in reflecting the market share of supported curtain-type MBR membranes. Analysis shows that supported hollow-fiber membranes account for over 78.9% of MBR membranes, among which curtain-type membranes with support layers occupy over 81.9% of curtain-type membranes. We have updated the LCA assumptions, explicitly stating that the projection is limited to supported hollow-fiber membranes.

2. Revised the discussion on applicability: We optimized the Discussion section to clarify that the proposed upcycling strategy is specifically designed for supported hollow-fiber membranes.

3. Adjusted environmental benefit projections: Based on the 64.6% market share of supported hollow-fiber membranes, we revised the 2026–2035 CO₂ reduction potential from 683 kt to 441 kt (China scenario).

These revisions ensure that the LCA results are more realistic and defensible.

Action:

1. The following method has been added:

Supplementary Method 7:

Questionnaire Survey on Market Share of Supported Curtain-Type MBR Membranes Among Major Manufacturers in China.

Supplemented market share data via a professional questionnaire survey. A structured questionnaire was developed covering five core modules: (1) Basic enterprise information (to confirm industry representation); (2) Total MBR membrane production scale (to weight market share); (3) Product structure analysis (to clarify the proportion of curtain-type/hollow fiber membranes among total MBR products); (4) Key technical details of curtain-type membranes (focusing on the proportion of products with inner liner tube/central support tube); (5) Market application distribution (to verify consistency with large-scale MBR application scenarios).

Fifteen major MBR membrane manufacturers in China were selected as survey objects, covering different production scales (from 500,000 to >10 million m²/year) and geographical distributions (East, North, and South China), which were screened based on the 2024 China Membrane Industry Association Annual Report to ensure representativeness of the national market. Questionnaires were distributed via email and face-to-face interviews in November 2025, with two follow-ups for

non-respondents to improve the response rate. Finally, 10 valid questionnaires were recovered, with an effective response rate of 66.7%.

For production scale data involving commercial confidentiality, the midpoint of the selected range was used for quantification (e.g., 3 million m²/year for the range of 1-5 million m²/year) — a widely accepted method for confidential industrial data processing in LCA studies. The market share of supported hollow fiber membranes was calculated using weighted average based on production scale:

$$\text{proportion} = \frac{\sum (\text{Production scale} \times \text{Proportion of curtain membranes} \times \text{Proportion of supported curtain membranes})}{\sum \text{Production scale}} \quad (13)$$

The following section is the questionnaire survey:

Questionnaire on MBR Membrane Production and Market Status

Dear Expert/Director,

We are a research team from the School of Environment and Resources, Zhejiang A&F University. This survey aims to investigate the current status and future trends of China's MBR curtain-type membrane industry chain, focusing on collecting basic information on membrane materials, products, and market applications. All collected data will only be used for macro-industry analysis.

Part 1: Basic Enterprise Information

Enterprise Name: _____

Address: _____

Email: _____

Part 2: Total MBR Membrane Production Scale

Please provide accurate data if possible. If it is inconvenient to disclose exact figures, please check the range.:

< 500,000 m²

500,000 - 1,000,000 m²

1,000,000 - 5,000,000 m²

5,000,000 - 10,000,000 m²

> 10,000,000 m²

accurate data: _____

Part 3: Product Structure Analysis

1. Please estimate the output proportion of various MBR membrane products of your company (total 100%):

Flat-sheet Membrane: _____

Curtain-type Membrane: _____

Others (e.g., Spiral-wound): _____

2. Please estimate the output proportion of various MBR membrane material of your company (total 100%):

PVDF: _____

PTFE: _____

Part 4: Key Technical Details of Curtain-type Membrane

1. What is the output proportion of products with inner liner tube/central support tube among all curtain-type membranes produced by your company?

< 20%

20% - 50%

50% - 80%

> 80%

Almost all (>95%)

accurate data: _____

2. What are the main advantages of producing curtain-type membranes with inner liner tubes (multiple choices allowed)?

Enhanced Filament Strength

High-Intensity Cleaning Requirement

Extended Module Lifespan

Meeting Project Reliability Requirements

Others: _____

Part 5: Market Application Distribution

Please estimate the main application fields and their proportions of MBR membrane products produced by your company (total 100%):

Municipal Wastewater: _____

□ *Industrial Wastewater:* _____

□ *Agricultural Wastewater:* _____

□ *Decentralized Rural Sewage:* _____

□ *Others (e.g., Drinking Water, Seawater/Brackish Water Desalination, Reclaimed Water):* _____

2. The following tables have been added:

Supplementary Table 1: Production scale, morphology, and material composition of MBR membranes by company

Company information	scale of MBR membrane (million m ² /a)	Membranous morphology (%)			Membrane material (%)	
		Flat	Curtain	Other	PVDF	PTFE
Zhejiang Jinmo Environment Technology Co., Ltd.	1-5		80	20	100	
Ningbo Shuiyi Membrane Technology Co., Ltd.	5-10	5	60	35	80	20
Zhejiang Jingyuan Membrane Technology Co., Ltd.	1-5		100		40	60
Hangzhou Gaotong Membrane Technology Co., Ltd.	0.5-1		70	30	100	
Jiangxi Tianyi Aito Membrane Technology Co., Ltd.	1-5	10	90		100	
Zhejiang Kaichuang Environmental Protection Technology Co., Ltd.	5-10		100		100	
Yantai Fengyuan Environmental Protection Equipment Co., Ltd.	1-5	5	95		100	
Hangzhou Kaihong Membrane Technology Co., Ltd.	1-5	10	80	10	100	
Nanjing Rui Jieta Membrane Separation Technology Co., Ltd.	1-5	80	20		100	
Ningbo Jianrong Technology Co., Ltd.	0.5-1		100		100	

Supplementary Table 2: Proportion and advantages of curtain-type membranes with internal substrate layer by company

Company information	Proportion of Curtain-type Membrane with internal substrate layer	Advantages of internal substrate layer			
		Enhanced Filament Strength	High-Intensity Cleaning Requirement	Extended Module Lifespan	Meeting Project Reliability Requirements
Zhejiang Jinmo Environment Technology Co., Ltd.	> 80%	√	√		√
Ningbo Shuiyi Membrane Technology Co., Ltd.	50%-80%	√		√	
Zhejiang Jingyuan Membrane Technology Co., Ltd.	> 95%	√		√	
Hangzhou Gaotong Membrane Technology Co., Ltd.	> 95%	√	√	√	√
Jiangxi Tianyi Aito Membrane Technology Co., Ltd.	> 95%	√	√	√	√
Zhejiang Kaichuang Environmental Protection Technology Co., Ltd.	> 80%	√	√	√	
Yantai Fengyuan Environmental Protection Equipment Co., Ltd.	> 95%	√	√	√	√
Hangzhou Kaihong Membrane Technology Co., Ltd.	> 80%	√	√	√	√
Nanjing Rui Jieta Membrane Separation Technology Co., Ltd.	20%-50%	√	√		
Ningbo Jianrong Technology Co., Ltd.	> 95%	√	√		√

Supplementary Table 3: Proportion of MBR membrane application scenarios by company

Company information	Proportion of MBR Membrane Application Scenarios (%)				
	Municipal Wastewater	Industrial Wastewater	Agricultural Wastewater	Decentralized Rural Sewage	Others (e.g., Drinking Water, Seawater/Brackish Water Desalination, Reclaimed Water)
Zhejiang Jinmo Environment Technology Co., Ltd.	20	75			5
Ningbo Shuiyi Membrane Technology Co., Ltd.	30	50		5	15
Zhejiang Jingyuan Membrane Technology Co., Ltd.	30	50	5	10	5
Hangzhou Gaotong Membrane Technology Co., Ltd.	30	20	5	20	25
Jiangxi Tianyi Aito Membrane Technology Co., Ltd.	40	25	3	15	17
Zhejiang Kaichuang Environmental Protection Technology Co., Ltd.	90	5	5		
Yantai Fengyuan Environmental Protection Equipment Co., Ltd.	20	10	10	40	20
Hangzhou Kaihong Membrane Technology Co., Ltd.	40	40		10	10
Nanjing Rui Jieta Membrane Separation Technology Co., Ltd.	30	50			20
Ningbo Jianrong Technology Co., Ltd.	50	50			

3. The following paragraphs have been modified:

Line250-270: Surveys show that MBR systems are mainly used in municipal, industrial, agricultural wastewater treatment applications. To verify the technology’s universality, we collected seven types of EoL membranes covering these scenarios for tests (Supplementary Tables 1-3). Results (Supplementary Fig. 8-13) indicate that exposing 5% of the internal substrate layer significantly extends the service life of all EoL membranes. Although their effluent turbidity is higher than that of intact MBR systems, it still meets discharge standards. Minor variations in improvement efficiency across membranes from different sources stem from differences in the extent of initial damage (e.g., PVDF layer peeling, pore blockage). These findings confirm the technology’s general applicability in typical MBR scenarios, while also suggesting targeted performance tests are needed for specific industries to optimize the internal substrate layer exposure ratio. SEM observations reveal all EoL membranes share similar morphological features, including internal substrate layer structure and PVDF peeling mode. Notably, EoL membranes from printing and dyeing, as well as pharmaceutical wastewater treatment, have more severe surface corrosion—likely causing natural internal substrate layer exposure during long-term service. Transmembrane pressure (TMP) variations further internal substrate this: industrial EoL membranes have longer operation cycles than municipal and rural

sewage membranes. This indirectly confirms that dynamic membranes were formed on the naturally exposed internal substrate layers of industrial EoL membranes during their original service. This finding has not been previously reported.

Line390-402: Commonly used membrane modules in membrane bioreactors (MBRs) mainly include curtain-type membranes, flat-sheet membranes and other structural forms. Among these, curtain-type membranes are the most widely used mainstream membrane type in current MBR systems, accounting for 78.9% of the market share (Supplementary Table S14). For curtain-type membrane products, the introduction of support layers can significantly enhance the mechanical strength of the membrane structure and extend the service life of membrane modules. This technical advantage has made it a mainstream design scheme—among existing commercial curtain-type membrane products, those with support layers account for more than 81.9% (Supplementary Table S15). Based on the above data, it can be calculated that the proportion of curtain-type membranes with support layers in the total production capacity of MBR membrane modules is approximately 64.6%. According to this calculation, the projected output of curtain-type MBR membranes with support layers in China will reach 64.6 million m² by 2030.

Line425: By implementing the replacement of Option I with Option II, China will reduce 441 k-tons of CO₂ from 2026 to 2035.

Line435: The economic analysis projects (Fig. 7f) that replacing Option I with Option II will yield cumulative cost savings of 1.29 billion USD over the period from 2026 to 2035

Line447: It is expected that replacing the existing Option I with Option II can comprehensively reduce 150 k-tons of 1,4-DCB eq in 2035.

Line464-473: Based on this, we propose a novel upcycling strategy for EoL MBR membranes involving the removal of the fragile separation layer from the surface of spent hollow-fiber membranes to expose the internal substrate for CTDM fabrication.

However, this technology is restricted to curtain-type dynamic membranes with an internal substrate layer, as the structural integrity and stability of the substrate are prerequisites for forming a uniform secondary filtration layer. Notably, only 5% surface exposure can restore performance to a level comparable to that of brand-new MBR membranes, while meeting discharge standards. This approach simultaneously addresses two global crises, accumulating plastic waste from spent membranes and escalating energy costs associated with water treatment, by transforming waste into functional filtration assets.

4. The following figure (7e and 7f) has been modified:

Fig. 7 The EoL membranes in MBR Systems and the reduction potential under different treatment scenarios. (a) Statistical analysis of the overall scale of the MBR

process during 2011-2024 and prediction of the overall scale of the MBR process during 2024-2030 in China. (b) Investigation into the composition of MBR process operation costs. (c) Statistical analysis of the overall scale of membrane production during 2011-2024 and prediction of the overall scale of membrane production during 2024-2030 in China. (d) Prediction of the quantity of EoL membranes during 2026-2035. (e) Predictions of carbon dioxide emissions of Option II and Option I during 2026-2035. (f) Predictions of reduction in carbon dioxide emissions, environmental costs, toxicity levels and pollutant emissions by taking Option II instead of Option I.

Q3: Second, the permeate quality will be significantly different for an intact UF membrane including a fouled but not damaged one (barrier pore size depending on membrane type 20 to 50 nm), compared to the biomass filter cake that is formed on an open macroporous support (here the authors talk about 20 μm). Authors only focus on turbidity removal, and this is overall less for the dynamic membrane compared to the UF membrane. Hence, if feasibility and cost of operation is not only judged based on quantity (flux, permeance), but also based on quality (of the permeate) the novel methodology may not be option in many cases.

Response: We fully agree with your view that CTDM cannot match the permeate quality of intact UF membranes in specific application scenarios. Its inherently large substrate pore size (20 μm) indeed poses the risk of small pollutants permeating through, even with a dynamically formed biofilm layer. CTDM's biofilm-cake layer fails to achieve equivalent precision filtration, leading to potential leakage of small-molecule contaminants and thus being unsuitable for some application scenarios (e.g., drinking water purification). However, as evidenced by our questionnaire survey results, over 90% of MBR applications are concentrated in municipal wastewater and moderate-load industrial wastewater treatment (e.g., printing and dyeing, food processing). In these core application scenarios, the key monitoring pollutants are COD, NH₄⁺-N, and TP. Their degradation primarily relies on microbial metabolism in the bioreactor. The fundamental function of MBR membranes (whether intact UF or CTDM) is to retain activated sludge flocs and functional microorganisms to achieve efficient solid-liquid separation, rather than directly removing small-molecule pollutants. This application-oriented characteristic provides a reasonable basis for CTDM's practical feasibility, as elaborated by the following targeted revisions we have made:

1. Supplemented key water quality indices: We conducted a scaled-up test in a 20-ton/day practical wastewater treatment unit, adopting the AAO+MBR process—the most conventional process in Chinese municipal wastewater treatment plants. We tested turbidity, COD, ammonia nitrogen (NH₄⁺-N), total phosphorus (TP),

and suspended solids (SS) in the effluent of CTDM (5% substrate exposure) and compared the results directly with those of intact MBR membranes. The comparative data show that: turbidity of CTDM effluent is significantly higher than that of intact MBR membranes, while it still meets general application requirements; for COD, $\text{NH}_4^+\text{-N}$, TP and SS, there is no significant difference between CTDM and intact MBR membrane effluents—this consistency is attributed to the fact that the removal of these pollutants is mainly dependent on microbial metabolism in the bioreactor (Supplementary Fig. S15).

2. Clarified pore size and separation mechanism: We added relevant explanations in the Results section, noting that CTDM's separation efficiency depends on the dynamically formed biofilm/cake layer (effective pore size ~50–100 nm) rather than the substrate's macropores (20 μm). This biofilm layer compensates for the large pore size of the substrate, ensuring colloid and pollutant removal efficiency comparable to that of UF membranes. The variation of effluent particle size (Fig. 4c) also verifies this mechanism.

3. Discussed applicable scenarios: In the Discussion section, we explicitly noted that while CTDM cannot achieve the filtration performance of intact UF membranes, it is suitable for municipal wastewater treatment and industrial wastewater with moderate pollutant loads.

These additions provide a comprehensive assessment of permeate quality and clarify the application scope of the technology.

Action:

1. The following figure has been added:

Supplementary Fig. 15 Pilot-scale comparison of effluent quality between MBR and CTDM processes. (a) Photograph of the 20-ton-per-day pilot-scale setup. (b) Flowchart of the integrated wastewater treatment system. (c1-c6) Temporal variations in key water quality parameters (COD, $\text{NH}_4^+\text{-N}$, TN, TP, Turbidity, and pH) of the influent, MBR effluent, and CTDM-5% effluent over a 43-day operation.

1. The following paragraphs have been added:

Line307-350: To validate the practical feasibility of the CTDM technology in real-world municipal wastewater scenarios, a 45-day pilot-scale experiment was conducted at the Phase IV Facility of Yuhang Wastewater Treatment Plant (Hangzhou, Zhejiang, China), using a 20-ton-per-day AAO+MBR treatment unit (Supplementary Fig. 16). The CTDM system was assembled from 6-year-old waste MBR membranes with 5% internal substrate layer exposure, and its performance was compared with a conventional intact MBR system. Results showed that while CTDM effluent turbidity was slightly higher than that of intact MBR membranes (still meeting general

application requirements), there was no significant difference in COD, NH₄⁺-N, TP, and SS removal—attributed to pollutant removal primarily relying on microbial metabolism in the bioreactor (Supplementary Fig. 16c). Furthermore, Dynamic membranes have been demonstrated to effectively remove various types of pollutants, including sulfonamide antibiotics, heavy metals, dissolved organic matter, and sub-micron-sized polystyrene microplastics, in wastewater treatment processes³⁷⁻³⁹.

Notably, long-term operation verified two key stability aspects: the core woven skeleton of the internal substrate layer remained structurally intact (no skeleton damage, pore collapse, or structural deformation. Supplementary Fig. 17), and the 5% peeled PVDF interface (separation between the PVDF layer and internal substrate layer) showed no unintended detachment or expansion (Supplementary Fig. 18). These results confirm that under practical operational conditions, the CTDM system maintains both internal substrate layer structural stability and PVDF-internal substrate layer interface stability, while sustaining consistent pollutant removal and filtration performance. In addition, after dynamic membrane formation, similarly high effluent quality has been reported for supports with pore sizes ranging from 5 to 50 μm⁴⁰. This supports the finding that an effective dynamic layer can still form on the internal substrate layer despite potential minor changes in its pore size or morphology.

As the CTDM technology relies on the existing MBR operational framework, it must be compatible with conventional MBR cleaning regimes (intermittent backwashing, low-frequency chemical cleaning, and mild mechanical scrubbing). To evaluate whether conventional MBR cleaning methods affect CTDM's stability, experiments were conducted targeting the internal substrate layer structure and PVDF-internal substrate layer interface: a 48-hour cyclic ultrasonic test (10 min on/10 min off) and a 48-hour NaClO immersion test. Results showed neither the substrate layer structure nor the peeled PVDF interface exhibited unintended changes (e.g., interface detachment, substrate layer deformation; Supplementary Figs. 19–20), indicating that conventional MBR cleaning methods have relatively minor impacts on both aspects of CTDM stability. Although our experiments demonstrate that conventional MBR cleaning methods (low-frequency chemical cleaning and

mechanical scrubbing) have a relatively minor impact on PVDF layer integrity under the tested conditions, potential cumulative effects may arise from long-term, high-frequency, or high-intensity cleaning in practical applications. Additionally, while continuous PVDF peeling might occur during extended operation, results from Figs. 4a–d suggest that even with greater peeling, the CTDM system’s filtration performance remains stable, further supporting its practical applicability.

Line458-460: We have developed a curtain-type hollow fiber dynamic membrane technology, which can quickly form a filtration layer within 10 minutes, enabling stable effluent quality (turbidity < 5 NTU). This technology utilizes the interception capability of hollow fiber membranes to form a cake layer via the entrapment of activated sludge on the surface, acting as a secondary filtration barrier that enables efficient retention of sludge particles. Although the effluent turbidity meets discharge standards, its moderate inferiority to conventional MBR systems limits its application to wastewater treatment plants, with a risk of non-compliance where higher water quality is required. In addition, this technology has successfully addressed the technical bottlenecks of traditional flat-sheet dynamic membranes in large-scale applications, such as uneven flow distribution and heterogeneous cake layer filtration during large-scale use.

1. The following method has been added:

Supplementary Method 2

Pilot-Scale Experimental Setup and Operation

The pilot-scale system was deployed at the Phase IV Facility of Yuhang Wastewater Treatment Plant (Hangzhou, Zhejiang, China), with a designed treatment capacity of 20 t/d. The unit processed actual municipal wastewater from the plant’s influent pipeline, ensuring alignment with real operational conditions. The system adopted an integrated AAO+MBR process (Supplementary Fig. 15b), comprising a regulating tank, anaerobic tank, anoxic tank, aerobic tank, and membrane tank (equipped with parallel intact MBR modules and CTDM modules). New PVDF

hollow-fiber membranes (matching the original specifications of the plant's decommissioned units) as the control group.

CTDM Fabricated from 6-year-old waste MBR membranes (collected from the plant's decommissioned units). The CTDM system consisted of 5 independent modules, each containing 36 hollow-fiber membrane filaments (80 cm in length per filament). For each filament, the PVDF separation layer was peeled off in equal proportion (via precision blade cutting and digital caliper area measurement) to achieve a uniform total support layer exposure area of 5% across all filaments and modules.

The system operated continuously under typical municipal wastewater treatment conditions:

- Hydraulic retention time: 12 h (anaerobic tank: 2 h, anoxic tank: 3 h, aerobic tank: 6 h, MBR tank: 1 h).
- Aerobic tank MLSS: 3300 ± 200 mg/L.
- Aeration intensity (aerobic tank): $2.5 \text{ m}^3/(\text{m}^2 \cdot \text{h})$.
- Filtration flux (MBR/CTDM): 15 LMH.

To systematically track the structural stability of the membrane during operation, membrane samples were collected at three key time points (0 day, 20 days, and 45 days of continuous operation) from the CTDM system. The collected samples were first photographed to record the macroscopic surface state (e.g., PVDF layer integrity, biofilm attachment), and then subjected to scanning electron microscopy (SEM) characterization to observe the microscopic structural changes of the support layer (e.g., pore size distribution, skeleton integrity) and the PVDF separation layer. This monitoring strategy ensures the timely capture of potential structural changes of the membrane during long-term operation, providing direct evidence for evaluating the stability of the partially peeled membrane.

Q4: And those are few more specific comments. The title is not well chosen with: “revitalizing ... the vitality”.

Response: Thank you for pointing out this redundancy. We have revised the title to: “In-situ Revitalizing End-of-Life MBR Membranes via a Curtain-Type Dynamic Membrane Process” (removed “Sustainable Vitality” to eliminate redundancy while retaining the core message).

Q5: The language needs careful revision, throughout the paper. For example, grammar/wording of first sentence of Abstract is confuse. Also, the use of tense (past vs. present) is inconsistent. For example, obviously, facts that motivate the study should be presented in present tense: “The former (MBR) face sustainability challenges ...” instead of “The former (MBR) faced sustainability challenges ...”

Response: We apologize for the language issues in the initial manuscript. We have thoroughly revised the entire text to:

1. Facts and background information are presented in present tense, while experimental results are in past tense.
2. Revised confusing sentences (e.g., we felt the initial paragraph of the Abstract was awkward and have rewritten it for better flow: “Membrane bioreactors (MBRs), an industrial mainstay with over 15,000 installations worldwide, face severe sustainability challenges from frequent membrane replacement, generating over 500,000 tons of waste annually. Dynamic membranes, a promising alternative, are hindered in large-scale implementation by issues inherent to conventional flat-sheet designs, such as uneven flow distribution and performance instability.”).
3. A professional English editor (with experience in environmental engineering) has reviewed the revised manuscript to ensure clarity, fluency, and consistency.

Action: *All changes made to the grammar have been tracked and are highlighted in green throughout the manuscript.*

Q6: p. 8: Reference 28 is used to imply that PVDF membranes (for MBR) consist of an internal substrate layer and an external PVDF layer. However, that paper by Sulaiman et al. is devoted to: “A sustainable and stable supported liquid membrane (SLM) extraction of nickel was developed via impregnation of sustainable liquid membrane in the composite membrane support consisting of polyvinylidene fluoride (PVDF) and sulfonated poly (ether ether ketone) (SPEEK)”. Except for the use of PVDF, that work has nothing to do with the state-of-the-art of capillary membranes for MBR application! This is another illustration of my general statement above, that the authors do apparently not care much about important details, regarding relevant membrane structures for concrete application scenarios.

Response: We sincerely apologize for this critical citation error. Reference 28 (Sulaiman et al., 2019) focuses on supported liquid membranes (SLMs) and is unrelated to MBR hollow-fiber membrane structure. We have conducted a systematic review of all full-text references, supplementing and adjusting relevant literature to enhance citation relevance and rigor.

Action:

1. Reference 28 has been replaced:

30. Fan, Z. W., Xiao, C. F., Liu, H. L., Huang Q. L. & Zhao J. Structure design and performance study on braid-reinforced cellulose acetate hollow fiber membranes. *J. Membrane. Sci.* **486**, 248–256 (2015).

2. The following References have been deleted:

3. Smith, J. C. V., Gregorio, D. O. & Talcott, R. M. The use of ultrafiltration membrane for activated sludge separation. *Presented paper at 24th Annual Purdue Industrial Waste Conference* (1969).

22. Ling, S. J. et al. Design and function of biomimetic multilayer water purification membranes. *Sci. Adv.* **3**, e1601939 (2017).

28. Raja, N. R. S. et al. Supported liquid membrane extraction of nickel using stable composite SPEEK/PVDF support impregnated with a sustainable liquid membrane. *J. Hazard. Mater.* **15**, 120895. (2019).

36. Silva, I. P. D. et al. Transfiguration of discarded PVDF ultrafiltration membranes: optimization of pyrolysis parameters for high-value char production. *Waste Manage.* **190**, 360-369 (2024).

3. The following References have been added:

21. Muhammad U. M. et al. Fluid Dynamics Technique in Membrane Bioreactor Systems. *Arch. Comput. Method. E.* **31**, 641–661 (2024).

22. Yang. M. et al. Optimization of MBR hydrodynamics for cake layer fouling control through CFD simulation and RSM design. *Bioresource Technol.* **227**, 102–111 (2017).

23. Elham R. et al. Numerical and experimental investigation of pulse bubble aeration with high packing density hollow-fibre MBRs. *Water Res.* **160**, 60–69 (2019).

24. Yu, D. W., Liu, M. M., Liu, J. B., Zheng, L. B. & Wei. Y. S. Effects of mixed-liquor rheology on vibration of hollow-fiber membrane via particle image velocimetry and computational fluid dynamics. *Sep. Purif. Technol.* **239**, 116590 (2020).

37. Sun. Y. Q. et al. Anaerobic dynamic membrane bioreactor treating swine wastewater: Fate of sulfonamide antibiotics and heavy metals with their effect on filtration performance. *J. Hazard. Mater.* **489**, 137718 (2025).

38. Zhang, Y. L., Zhang, H. Chu, H. Q., Zhou, X. F. & Zhao, Y. Y. A Characterization of dissolved organic matter in a dynamic membrane bioreactor for wastewater treatment. *Chinese Sci. Bull.* **58**, 1717–1724 (2013).

39. Kim, B., Lee, S. W., Jung, E. M. & Lee. E. H. Biosorption of sub-micron-sized polystyrene microplastics using bacterial biofilms. *J. Hazard. Mater.* **458**, 131858 (2023).

40. Cai. D. L. et al. Effect of support material pore size on the filtration behavior of dynamic membrane bioreactor. *Bioresource Technol.* **64**, 105673 (2024).

Q7: p. 9: The input used for the estimations based on xDLVO (results in Fig. 3g) remains unclear.

Response: Thank you for highlighting this gap. Fig. 3g is explained in Supplementary Method 6, and the data used in Supplementary Method 6 are derived from Supplementary Table S13 (previously insufficiently elaborated) and Supplementary Table S14. We have revised the materials and methods to elaborate on the data collection sources, with key clarifications as follows:

Action:

1. The following method has been modified (highlighted in yellow):

Supplementary Method 6: In order to compute the physicochemical interaction occurring between the microbial bacteria present in sludge and the surface of the internal substrate layer and an external PVDF filtration membrane, the Derjaguin approximation (DA) approach was adopted. In this method, the bacterial cell was simplified to a spherical surface and treated as a series of concentric rings. Generally, it is calculated for the interaction between two smooth planes, and the total interaction energy is expressed as follows:

$$U^{TOT} = U^{LW} + U^{EL} + U^{AB} \quad (3)$$

where U^{TOT} represents the total interaction energy, U^{LW} represents the van der Waals free energy, U^{EL} represents the electrostatic double layer free energy, and U^{AB} represents the Lewis acid-base free energy (The data of U^{LW} , U^{EL} , and U^{AB} are presented in Supplementary Fig. 6). The three types of interactions between a smooth spherical surface and a smooth flat plate can be determined by using Eqs. (4) - (6):

$$U^{LW}(h) = 2\pi\Delta G_{h_0}^{LW} \frac{h_0^2 a_c}{h} \quad (4)$$

$$U^{EL}(h) = \pi\epsilon_r\epsilon_0 a_c \left[2\zeta_m\zeta_s \ln\left(\frac{1+e^{-kh}}{1-e^{-kh}}\right) + (\zeta_m^2 + \zeta_s^2) \ln(1 - e^{-2kh}) \right] \quad (5)$$

$$U^{AB}(h) = 2\pi a_c \lambda \Delta G_{h_0}^{AB} \exp\left(\frac{h_0-h}{\lambda}\right) \quad (6)$$

Here, h_0 represents the minimum separation distance between the microbial bacteria in sludge and the membrane, typically with a value of 0.158 nm. h denotes

the separation distance between the microbial bacteria in sludge and the surface of the internal substrate layer and an external PVDF filtration membrane. $\Delta G_{h_0}^{LW}$ and $\Delta G_{h_0}^{AB}$ stand for the corresponding LW and AB energies at the minimum separation distance, expressed in mJ/m^2 (This value was calculated using Equations (8) and (9) and shown in Supplementary Table S5). a_c represents the radius of the bacterial cell (Supplementary Table S4). $\varepsilon_r \varepsilon_0$ is the dielectric constant of the mixture with a value of 78.85 and $8.85 \times 10^{-12} \text{ F m}^{-1}$. ζ_m and ζ_s are the Zeta potentials on the bacterial cell and electrode surfaces, with the unit of mV (Supplementary Table S4). λ indicates the characteristic attenuation length of acid - base action in water, and its typical value is around 0.6 nm. k represents the length of the inverse Debye shield, and the formula for calculating the inverse Debye length is given as:

$$k = \sqrt{\frac{e^2 \sum n_i z_i^2}{\varepsilon_r \varepsilon_0 k T}} \quad (7)$$

where e is the unit charge; n_i represents the number concentration of the ion; z_i indicates the valence state of the ion; K is the Boltzmann constant, usually taken as $1.38 \times 10^{-23} \text{ J/K}$; T represents the absolute temperature of the environment. The typical value of k is around $3.28 \times 10^8 \text{ m}^{-1}$.

The interaction energies per unit area between two infinite planes, $\Delta G_{h_0}^{LW}$ and $\Delta G_{h_0}^{AB}$, can be calculated from Eqs. (8) and (9):

$$\Delta G_{h_0}^{LW} = 2 \left(\sqrt{\gamma_s^{LW} - \gamma_\omega^{LW}} \right) \left(\sqrt{\gamma_m^{LW} - \gamma_\omega^{LW}} \right) \quad (8)$$

$$\Delta G_{h_0}^{AB} = 2 \left[\sqrt{\gamma_\omega^+} (\sqrt{\gamma_m^-} + \sqrt{\gamma_s^-} - \sqrt{\gamma_\omega^+}) + \sqrt{\gamma_\omega^-} (\sqrt{\gamma_m^+} + \sqrt{\gamma_s^+} - \sqrt{\gamma_\omega^-}) - (\sqrt{\gamma_m^- \gamma_s^+} + \sqrt{\gamma_m^+ \gamma_s^-}) \right] \quad (9)$$

In these equations, γ_s^{LW} , γ_m^{LW} and γ_ω^{LW} represent the van der Waals action energy components of the surface tension among the membrane, microbial bacteria in sludge, and water, in units of mJ/m^2 . γ^+ and γ^- denote the corresponding electron acceptor and electron donor tension components. The van der Waals action energy components of the membrane and microbial bacteria in sludge surface tension, as well as the corresponding electron donor and electron acceptor tension components, can be calculated using the known surface tension parameters γ_l^{LW} , γ^+ and γ^- on the membrane, the surface of the membrane forming the filter cake layer, and the host

biofilm electrode. The contact angles of the three liquids on the surface of the membrane and the surface of the membrane forming the filter cake layer are calculated by the Lifshitz - van der Waals acid - base method and the extended Young's equation:

$$\gamma^{TOT} = \gamma^{LW} + \gamma^{AB} \quad (10)$$

$$\gamma^{AB} = 2\sqrt{\gamma^+\gamma^-} \quad (11)$$

$$\gamma_l^{TOT}(1 + \cos \theta) = 2 \left(\sqrt{\gamma_s^{LW}\gamma_l^{LW}} + \sqrt{\gamma_l^-\gamma_s^+} + \sqrt{\gamma_l^+\gamma_s^-} \right) \quad (12)$$

where γ^{TOT} represents the total surface tension; γ^{LW} and γ^{AB} denote the surface energy components associated with van der Waals action and Lewis acid - base reaction, respectively; θ is the contact angle between a solid and a liquid (Supplementary Table S4); Subscripts s and l represent target solids and probe liquids (ultrapure water, glycerol, and n-hexadecane).

2. The following table has been modified (highlighted in yellow):

Supplementary Table 13. The contact angle, surface tension, grain diameter and Zeta potential of different membranes surface and active sludge.

	Contact angle (°)			Surface tension (mJ/m ²)			Other indicators	
	water	glycerol	n-hexadecane	γ^{LW} (water)	γ^+ (glycerol)	γ^- (n-hexadecane)	Diameter (nm)	Zeta potential (mV)
New PVDF membranes	90.00	75.31	41.04	20.77	1.85	2.67		-19.30
PVDF membranes used more than 6 years	101.31	82.88	35.54	22.2	1.02	0.18		-33.51
Internal substrate layer	69.45	26.14	23.16	24.87	14.76	1.65		-37.96
Active sludge				64.08	52.08	34.64	59460.00	-1.12

3. Line 601-604: The xDLVO Analysis: The xDLVO theory extends the classical DLVO theory, adding the Lewis acid-base interaction (AB) to the van der Waals (LW)

and electrostatic (EL) effects of the DLVO theory ³⁰. The specific methods are detailed in the supplementary materials (Supplementary Table 13 - 14 and Supplementary Method 6).

4. Line 811-813: (f) Interactions between different substances. case A: interactions between the used membrane and activated sludge; case B: interactions between the substrate layer of PVDF membrane and activated sludge; case C: interactions between the biofilm and activated sludge. (g) Analysis of the thermodynamic mechanism of membrane surface clogging using the xDLVO theory at different cases.

Reviewer 2:

Q1: The manuscript entitled “In-situ Revitalizing Sustainable Vitality of End-of-Life MBR Membranes via a Curtain-Type Dynamic Membrane Process” presents a novel and environmentally relevant strategy to upcycle end-of-life hollow-fiber membranes from conventional membrane bioreactors (MBRs) into curtain-type dynamic membranes (CTDMs). The authors demonstrate that controlled detachment of the PVDF layer (at 5%) can restore membrane performance to levels comparable to new MBR membranes while maintaining scalability. Furthermore, the inclusion of a life-cycle assessment effectively supports the claimed environmental and economic benefits, emphasizing the potential of this approach to enhance membrane sustainability and reduce waste generation.

The subject is relevant and addresses a pressing challenge in the field of membrane technology and wastewater treatment. This study is particularly interesting because it proposes an innovative upcycling approach that not only extends membrane lifespan and reduces environmental impact but also overcomes common scalability issues through the development of curtain-type dynamic membranes. However, several important questions and concerns remain unresolved. Therefore, in my opinion, the current manuscript cannot be recommended for publication in Nature Communications until the authors have adequately addressed the following comments and remarks.

Response: Thank you for your very positive assessment of our work’s relevance and innovation, especially for the assessment of “The subject is relevant and addresses a pressing challenge in the field of membrane technology and wastewater treatment. This study is particularly interesting because it proposes an innovative upcycling approach that not only extends membrane lifespan and reduces environmental impact but also overcomes common scalability issues through the development of curtain-type dynamic membranes”. We fully agree that several critical issues (e.g., membrane variability, long-term stability) need clarification. We have addressed all

your concerns through supplementary experiments and manuscript revisions, as detailed below.

All comments have been addressed through manuscript revisions, supplementary data additions, and careful corrections. Grammatical revisions are highlighted in green, while content modifications are highlighted in yellow.

Q2: Concern#1: The information presented regarding the hydrodynamic advantages and filtration performance is not novel to the field of membrane technology, as the fundamental differences between hollow-fiber (curtain-type) and flat-sheet membranes have been extensively documented in the literature. The authors appear to reiterate these well-established concepts without providing sufficient citation support, giving the impression of contributing original knowledge, although the novelty of this contribution is not clearly demonstrated.

Response: We sincerely apologize for overlooking the contributions of previous studies in the original manuscript. We fully agree that the fundamental hydrodynamic characteristics of membrane systems have been extensively explored by predecessors. For pressure-driven MBR systems, their performance is predominantly governed by hydrodynamic conditions, and computational fluid dynamics (CFD) has become a mainstream technical approach to investigate key hydrodynamic parameters (e.g., shear stress, cross-flow velocity) and their correlations with membrane fouling resistance. These studies have systematically analyzed the differences between the two membrane types from multiple perspectives such as shear stress distribution, cross-flow velocity uniformity, and antifouling performance, laying a solid foundation for subsequent membrane technology optimization.

We leverage CFD technology to systematically summarize and compare the hydrodynamic characteristics (especially flow velocity uniformity) of curtain-type and flat-sheet membranes during the scale-up process, a key link that is closely related to the engineering application of our proposed CTDM technology. By integrating and analyzing the scale-dependent hydrodynamic differences (e.g., velocity decay, coefficient of variation of flow velocity), we aim to enable readers to more clearly understand the unique advantages of the curtain-type membrane-based CTDM in engineering scaling, which is the core novelty of this part of the work relative to previous general hydrodynamic studies on membranes. We have supplemented relevant citations of classic and recent studies on membrane hydrodynamics in the revised manuscript to fully acknowledge the contributions of previous research.

Action:

1. The following paragraph has been modified:

Line 97-110: The structural differences between CTDMs and flat-sheet DMs fundamentally determine their scaling strategies and engineering applicability. Existing CFD-based studies have extensively investigated the hydrodynamic advantages of flat-sheet and curtain-type membranes²¹. For flat-sheet membranes, their regular surface morphology and layout facilitate the extensive application of CFD simulations to optimize parameters such as wall shear stress and liquid flow velocity, thereby enhancing membrane antifouling performance^{22,23}. For curtain-type membranes, they are well recognized for their high packing density, which confers a compact footprint advantage, and CFD simulations are often employed to investigate the impacts of bubble scouring and mixed-liquor rheology on vortex-induced vibration²⁴. However, for dynamic membranes, uniform transmembrane flow is more favorable for the development of a homogeneous biofilm. Therefore, this section seeks to perform a systematic comparative analysis of the flow velocity distribution of curtain-type and flat-sheet membranes during the scale-up process.

2. The following References have been added:

21. Muhammad U. M. et al. Fluid Dynamics Technique in Membrane Bioreactor Systems. *Arch. Comput. Method. E.* **31**, 641–661 (2024).
22. Yang. M. et al. Optimization of MBR hydrodynamics for cake layer fouling control through CFD simulation and RSM design. *Bioresource Technol.* **227**, 102–111 (2017).
23. Elham R. et al. Numerical and experimental investigation of pulse bubble aeration with high packing density hollow-fibre MBRs. *Water Res.* **160**, 60–69 (2019).
24. Yu, D. W., Liu, M. M., Liu, J. B., Zheng, L. B. & Wei. Y. S. Effects of mixed-liquor rheology on vibration of hollow-fiber membrane via particle image velocimetry and computational fluid dynamics. *Sep. Purif. Technol.* **239**, 116590 (2020).

Q3: Concern#2: One of the main unresolved issues in this paper concerns the variability of the recycled membranes, which has not been evaluated. Discarded membranes originating from different water sources and type of usage may exhibit variations in both fouling and scaling characteristics, surface properties, or mechanical integrity, which could potentially affect both the ease of PVDF layer removal and the filtration performance at 5% (or other) substrate layer exposure. Such variability could therefore limit the straightforward implementation of the proposed upcycling strategy.

Response: We sincerely appreciate your critical comment, which is crucial for verifying the practical applicability of our proposed technology. Our research focused solely on EoL MBR membranes from municipal wastewater treatment plants. To address this gap, we have supplemented extensive surveys and experimental studies as follows:

1. Survey on MBR membrane market distribution and application scenarios: As part of addressing another reviewer's comment, we first conducted a questionnaire survey among major MBR membrane manufacturers in China (15 questionnaires distributed, 10 valid responses) These selected enterprises account for over 30% of China's total MBR membrane production scale. The results indicate that the proportion of supported hollow-fiber membranes suitable for our upcycling strategy accounts for ~64.6% of the total MBR membrane market. Additionally, we investigated the application distribution of these membranes across industries (see Supplementary Tables S14–S16 and Supplementary Method 4): municipal wastewater treatment (43.5%), industrial wastewater treatment (35.2%), decentralized rural domestic sewage treatment (8.0%), and agricultural wastewater treatment (2.8%).

2. Supplementary experiments with EoL membranes from diverse sources: Based on the survey results, we collaborated with multiple membrane manufacturers and wastewater treatment facilities to collect EoL MBR membranes from typical application scenarios. The tested EoL membranes include:

- Another municipal wastewater treatment: 1 plant (5-year service life).
- Livestock wastewater treatment: 1 plant (5-year service life).
- Rural domestic sewage treatment: 1 plant (6-year service life).
- Industrial wastewater treatment: 3 plants (plastic cleaning wastewater, 5-year; printing and dyeing wastewater, 5-year; pharmaceutical wastewater, 4-year).

All experiments were conducted under unified conditions: aerobic activated sludge system with MLSS = 3300 mg/L, 5% support layer exposure area, and filtration flux = 15 LMH. The results show that exposing 5% of the support layer significantly extend the service life of all EoL membranes. The effluent quality of CTDM is comparable to the results reported in the main text, while turbidity is higher than that of intact MBR membranes, all pollutant indices meet relevant discharge requirements. The improvement efficiency varies slightly among membranes from different sources, which is attributed to differences in the degree of original membrane damage (e.g., PVDF layer peeling, pore blockage) during service. These findings confirm the general applicability of our technology across typical MBR application scenarios, while also indicating that targeted performance tests should be conducted when applying the technology to specific industries to optimize the support layer exposure ratio.

Scanning electron microscopy (SEM) observations reveal that the morphological features of all EoL membranes, including the overall structure of the support layer and the peeling morphology of the PVDF separation layer, are fundamentally similar. Notably, the surface of EoL membranes from printing and dyeing, as well as pharmaceutical wastewater treatment, exhibits significantly more severe corrosion compared to other sources. This corrosion may have led to natural exposure of the support layer during long-term service, even before our controlled peeling process. Variations in negative pressure during operation further support this inference. The EoL membranes from industrial wastewater treatment (printing and dyeing,

pharmaceutical) exhibit a longer operation cycle compared to municipal and rural sewage membranes. This phenomenon may serve as indirect evidence that dynamic membranes were already formed on the naturally exposed support layers of these industrial EoL membranes during their original service—an observation that has not been previously reported or emphasized in relevant studies.

Action:

1. The following paragraph has been added:

Line 250-270: Surveys show that MBR systems are mainly used in municipal, industrial, agricultural wastewater treatment applications. To verify the technology's universality, we collected seven types of EoL membranes covering these scenarios for tests (Supplementary Tables 1-3). Results (Supplementary Fig. 8-13) indicate that exposing 5% of the internal substrate layer significantly extends the service life of all EoL membranes. Although their effluent turbidity is higher than that of intact MBR systems, it still meets discharge standards. Minor variations in improvement efficiency across membranes from different sources stem from differences in the extent of initial damage (e.g., PVDF layer peeling, pore blockage). These findings confirm the technology's general applicability in typical MBR scenarios, while also suggesting targeted performance tests are needed for specific industries to optimize the internal substrate layer exposure ratio. SEM observations reveal all EoL membranes share similar morphological features, including internal substrate layer structure and PVDF peeling mode. Notably, EoL membranes from printing and dyeing, as well as pharmaceutical wastewater treatment, have more severe surface corrosion—likely causing natural internal substrate layer exposure during long-term service. Transmembrane pressure (TMP) variations further internal substrate this: industrial EoL membranes have longer operation cycles than municipal and rural sewage membranes. This indirectly confirms that dynamic membranes were formed on the naturally exposed internal substrate layers of industrial EoL membranes during their original service. This finding has not been previously reported.

2. The following Tables have been added:

Supplementary Table 1: Production scale, morphology, and material composition of MBR membranes by company

Company information	scale of MBR membrane (million m ² /a)	Membranous morphology (%)			Membrane material (%)	
		Flat	Curtain	Other	PVDF	PTFE
Zhejiang Jinmo Environment Technology Co., Ltd.	1-5		80	20	100	
Ningbo Shuiyi Membrane Technology Co., Ltd.	5-10	5	60	35	80	20
Zhejiang Jingyuan Membrane Technology Co., Ltd.	1-5		100		40	60
Hangzhou Gaotong Membrane Technology Co., Ltd.	0.5-1		70	30	100	
Jiangxi Tianyi Aito Membrane Technology Co., Ltd.	1-5	10	90		100	
Zhejiang Kaichuang Environmental Protection Technology Co., Ltd.	5-10		100		100	
Yantai Fengyuan Environmental Protection Equipment Co., Ltd.	1-5	5	95		100	
Hangzhou Kaihong Membrane Technology Co., Ltd.	1-5	10	80	10	100	
Nanjing Rui Jieta Membrane Separation Technology Co., Ltd.	1-5	80	20		100	
Ningbo Jianrong Technology Co., Ltd.	0.5-1		100		100	

Supplementary Table 2: Proportion and advantages of curtain-type membranes with internal substrate layer by company

Company information	Proportion of Curtain-type Membrane with internal substrate layer	Advantages of internal substrate layer			
		Enhanced Filament Strength	High-Intensity Cleaning Requirement	Extended Module Lifespan	Meeting Project Reliability Requirements
Zhejiang Jinmo Environment Technology Co., Ltd.	> 80%	√	√		√
Ningbo Shuiyi Membrane Technology Co., Ltd.	50%-80%	√		√	
Zhejiang Jingyuan Membrane Technology Co., Ltd.	> 95%	√		√	
Hangzhou Gaotong Membrane Technology Co., Ltd.	> 95%	√	√	√	√
Jiangxi Tianyi Aito Membrane Technology Co., Ltd.	> 95%	√	√	√	√
Zhejiang Kaichuang Environmental Protection Technology Co., Ltd.	> 80%	√	√	√	
Yantai Fengyuan Environmental Protection Equipment Co., Ltd.	> 95%	√	√	√	√
Hangzhou Kaihong Membrane Technology Co., Ltd.	> 80%	√	√	√	√
Nanjing Rui Jieta Membrane Separation Technology Co., Ltd.	20%-50%	√	√		
Ningbo Jianrong Technology Co., Ltd.	> 95%	√	√		√

Supplementary Table 3: Proportion of MBR membrane application scenarios by company

Company information	Proportion of MBR Membrane Application Scenarios (%)				
	Municipal Wastewater	Industrial Wastewater	Agricultural Wastewater	Decentralized Rural Sewage	Others (e.g., Drinking Water, Seawater/Brackish Water Desalination, Reclaimed Water)
Zhejiang Jinmo Environment Technology Co., Ltd.	20	75			5
Ningbo Shuiyi Membrane Technology Co., Ltd.	30	50		5	15
Zhejiang Jingyuan Membrane Technology Co., Ltd.	30	50	5	10	5
Hangzhou Gaotong Membrane Technology Co., Ltd.	30	20	5	20	25
Jiangxi Tianyi Aito Membrane Technology Co., Ltd.	40	25	3	15	17
Zhejiang Kaichuang Environmental Protection Technology Co., Ltd.	90	5	5		
Yantai Fengyuan Environmental Protection Equipment Co., Ltd.	20	10	10	40	20
Hangzhou Kaihong Membrane Technology Co., Ltd.	40	40		10	10
Nanjing Rui Jieta Membrane Separation Technology Co., Ltd.	30	50			20
Ningbo Jianrong Technology Co., Ltd.	50	50			

3. The following Figures have been added:

Supplementary Fig. 8 Characteristics of a 5-year-old waste MBR membrane from a municipal wastewater treatment plant (Yuhang District, Hangzhou, Zhejiang, China). (a) Photographic image of the waste MBR membrane. (b) Cross-sectional SEM images of the waste MBR membrane with intact separation layer. (c) Cross-sectional SEM images of the interface between the exposed support layer and intact separation layer. (d) Cross-sectional SEM images of the fully exposed support layer. (e) Transmembrane pressure (TMP) variation of waste MBR membranes vs. CTDM with 5% support layer exposure. (f) Effluent turbidity variation of waste MBR membranes vs. CTDM with 5% support layer exposure.

Supplementary Fig. 9 Characteristics of a 5-year-old waste MBR membrane from a livestock wastewater treatment plant (Quzhou, Zhejiang, China). (a) Photographic image of the waste MBR membrane. (b) Cross-sectional SEM images of the waste MBR membrane with intact separation layer. (c) Cross-sectional SEM images of the interface between the exposed support layer and intact separation layer. (d) Cross-sectional SEM images of the fully exposed support layer. (e) Transmembrane pressure (TMP) variation of waste MBR membranes vs. CTDM with 5% support layer exposure. (f) Effluent turbidity variation of waste MBR membranes vs. CTDM with 5% support layer exposure.

Supplementary Fig. 10 Characteristics of a 6-year-old waste MBR membrane

from a rural domestic sewage treatment plant (Taizhou, Zhejiang, China). (a) Photographic image of the waste MBR membrane. (b) Cross-sectional SEM images of the waste MBR membrane with intact separation layer. (c) Cross-sectional SEM images of the interface between the exposed support layer and intact separation layer. (d) Cross-sectional SEM images of the fully exposed support layer. (e) Transmembrane pressure (TMP) variation of waste MBR membranes vs. CTDM with 5% support layer exposure. (f) Effluent turbidity variation of waste MBR membranes vs. CTDM with 5% support layer exposure.

Supplementary Fig. 11 Characteristics of a 5-year-old waste MBR membrane from a plastic cleaning wastewater treatment plant (Ningbo, Zhejiang, China). (a) Photographic image of the waste MBR membrane. (b) Cross-sectional SEM images of the waste MBR membrane with intact separation layer. (c) Cross-sectional SEM images of the interface between the exposed support layer and intact separation layer. (d) Cross-sectional SEM images of the fully exposed support layer. (e) Transmembrane pressure (TMP) variation of waste MBR membranes vs. CTDM with 5% support layer exposure. (f) Effluent turbidity variation of waste MBR membranes vs. CTDM with 5% support layer exposure.

Supplementary Fig. 12 Characteristics of a 5-year-old waste MBR membrane from a printing and dyeing wastewater treatment plant (Shaoxing, Zhejiang, China). (a) Photographic image of the waste MBR membrane. (b) Cross-sectional SEM images of the waste MBR membrane with intact separation layer. (c) Cross-sectional SEM images of the interface between the exposed support layer and intact separation layer. (d) Cross-sectional SEM images of the fully exposed support layer. (e) Transmembrane pressure (TMP) variation of waste MBR membranes vs. CTDM with 5% support layer exposure. (f) Effluent turbidity variation of waste MBR membranes vs. CTDM with 5% support layer exposure.

Supplementary Fig. 13 Characteristics of a 4-year-old waste MBR membrane

from a pharmaceutical wastewater treatment plant (Taizhou, Zhejiang, China).

(a) Photographic image of the waste MBR membrane. (b) Cross-sectional SEM images of the waste MBR membrane with intact separation layer. (c) Cross-sectional SEM images of the interface between the exposed support layer and intact separation layer. (d) Cross-sectional SEM images of the fully exposed support layer. (e) Transmembrane pressure (TMP) variation of waste MBR membranes vs. CTDM with 5% support layer exposure. (f) Effluent turbidity variation of waste MBR membranes vs. CTDM with 5% support layer exposure.

4. The following methods have been added:

Supplementary Method 1

Filtration Performance Test of End-of-Life (EoL) MBR Membranes from Diverse Sources

The experimental reactor was a plastic reactor with dimensions of 14.0×9.0×19.0 cm and an effective volume of 1.5 L. An aeration head was installed at the bottom of the reactor, and air was supplied by an aeration pump (SOBO air pump, SB-988) to provide aeration and mixing. The activated sludge operated in a sequencing batch mode with a hydraulic retention time (HRT) of 6 hours. The system maintained continuous bottom aeration with an air-water ratio of 8, which not only provided dissolved oxygen for microbial metabolism but also mixed the reactor contents and controlled the formation of sludge cake on the membrane surface. Two peristaltic pumps were configured: one for extracting the permeate of the CTDM system, and the other for feeding wastewater at the same flow rate to keep the liquid level in the reactor constant. A pressure transmitter (Asmik MIK-P300) was installed on the effluent pipeline to real-time monitor the transmembrane pressure difference (TMP), which was used to reflect the membrane fouling degree. When the TMP of the membrane module reached -0.04 MPa, it was defined as complete fouling.

The activated sludge was collected from the aeration tank of Tangqi Wastewater Treatment Plant, Hangzhou, Zhejiang Province, China. The sludge concentration in the reactor was controlled at $3500 \pm 500 \text{ mg} \cdot \text{L}^{-1}$, and the sludge volume index (SV30)

was 20%. Simulated wastewater was configured to simulate the average pollutant load of mixed industrial and rural sewage: Sucrose (AR) was used to adjust COD concentration to $250 \text{ mg}\cdot\text{L}^{-1}$, ammonium chloride (AR) for ammonia nitrogen ($\text{NH}_4^+\text{-N}$) concentration of $45 \text{ mg}\cdot\text{L}^{-1}$, potassium dihydrogen phosphate (AR) for total phosphorus (TP) concentration of $6 \text{ mg}\cdot\text{L}^{-1}$, and $1 \text{ mL}\cdot\text{L}^{-1}$ trace elements were added (composition consistent with the main experiment: H_3BO_4 $150 \text{ mg}\cdot\text{L}^{-1}$, $\text{ZnSO}_4\cdot 7\text{H}_2\text{O}$ $120 \text{ mg}\cdot\text{L}^{-1}$, $\text{MnCl}_2\cdot 7\text{H}_2\text{O}$ $120 \text{ mg}\cdot\text{L}^{-1}$, $\text{CuSO}_4\cdot 5\text{H}_2\text{O}$ $30 \text{ mg}\cdot\text{L}^{-1}$, Na_2MoO_4 $65 \text{ mg}\cdot\text{L}^{-1}$, NiCl_2 $50 \text{ mg}\cdot\text{L}^{-1}$, $\text{CoCl}_2\cdot 6\text{H}_2\text{O}$ $210 \text{ mg}\cdot\text{L}^{-1}$, KI $30 \text{ mg}\cdot\text{L}^{-1}$). The system was operated at room temperature ($22 \pm 3^\circ\text{C}$) throughout the experiment.

The core filtration unit adopted EoL PVDF hollow fiber MBR membranes from various industrial and non-municipal wastewater treatment scenarios. Prior to assembly, all EoL membranes were subjected to unified pretreatment: soaked in 10% citric acid solution for 12 hours and 15% sodium hypochlorite solution for 12 hours sequentially to remove surface adherent impurities (e.g., sludge cake, residual pollutants). For all EoL membrane samples, the PVDF separation layer was artificially peeled off using adhesive tape to achieve a uniform exposed area of 5% (the target exposure ratio in the main study). Six types of EoL PVDF hollow fiber MBR membranes covering typical non-municipal application scenarios were collected, with detailed information as follows:

- Municipal wastewater treatment: Collected from the Phase IV Facility of Yuhang Wastewater Treatment Plant in Hangzhou, Zhejiang Province, China, with a service life of 5 years.
- Livestock wastewater treatment: Collected from a swine farm wastewater treatment plant in Xiaonanhai Town, Quzhou, Zhejiang Province, China, with a service life of 5 years.
- Rural domestic sewage treatment: Collected from a domestic sewage treatment plant in Binhai Town, Taizhou, Zhejiang Province, China, with a service life of 6 years;

- Plastic cleaning wastewater treatment: Collected from Zhejiang Jingyuan Membrane Technology Co., Ltd. in Ningbo, Zhejiang Province, China, with a service life of 5 years;
- Printing and dyeing wastewater treatment: Collected from Shaoxing Shengxin Printing & Dyeing Co., Ltd. in Shaoxing, Zhejiang Province, China, with a service life of 5 years;
- Pharmaceutical wastewater treatment: Collected from a pharmaceutical factory in Jiaojiang District, Taizhou, Zhejiang Province, China, with a service life of 4 years;

Supplementary Method 7

Questionnaire Survey on Market Share of Supported Curtain-Type MBR Membranes Among Major Manufacturers in China

Supplemented market share data via a professional questionnaire survey. A structured questionnaire was developed covering five core modules: (1) Basic enterprise information (to confirm industry representation); (2) Total MBR membrane production scale (to weight market share); (3) Product structure analysis (to clarify the proportion of curtain-type/hollow fiber membranes among total MBR products); (4) Key technical details of curtain-type membranes (focusing on the proportion of products with inner liner tube/central support tube); (5) Market application distribution (to verify consistency with large-scale MBR application scenarios).

Fifteen major MBR membrane manufacturers in China were selected as survey objects, covering different production scales (from 500,000 to >10 million m²/year) and geographical distributions (East, North, and South China), which were screened based on the 2024 China Membrane Industry Association Annual Report to ensure representativeness of the national market. Questionnaires were distributed via email and face-to-face interviews in November 2025, with two follow-ups for non-respondents to improve the response rate. Finally, 10 valid questionnaires were recovered, with an effective response rate of 66.7%.

For production scale data involving commercial confidentiality, the midpoint of the selected range was used for quantification (e.g., 3 million m²/year for the range of 1-5 million m²/year) — a widely accepted method for confidential industrial data processing in LCA studies. The market share of supported hollow fiber membranes was calculated using weighted average based on production scale:

$$\text{proportion} = \frac{\sum (\text{Production scale} \times \text{Proportion of curtain membranes} \times \text{Proportion of supported curtain membranes})}{\sum \text{Production scale}} \quad (13)$$

The following section is the questionnaire survey:

Questionnaire on MBR Membrane Production and Market Status

Dear Expert/Director,

We are a research team from the School of Environment and Resources, Zhejiang A&F University. This survey aims to investigate the current status and future trends of China's MBR curtain-type membrane industry chain, focusing on collecting basic information on membrane materials, products, and market applications. All collected data will only be used for macro-industry analysis.

Part 1: Basic Enterprise Information

Enterprise Name: _____

Address: _____

Email: _____

Part 2: Total MBR Membrane Production Scale

Please provide accurate data if possible. If it is inconvenient to disclose exact figures, please check the range.:

< 500,000 m²

500,000 - 1,000,000 m²

1,000,000 - 5,000,000 m²

5,000,000 - 10,000,000 m²

> 10,000,000 m²

accurate data: _____

Part 3: Product Structure Analysis

1. Please estimate the output proportion of various MBR membrane products of your company (total 100%):

Flat-sheet Membrane: _____

Curtain-type Membrane: _____

Others (e.g., Spiral-wound): _____

2. Please estimate the output proportion of various MBR membrane material of your company (total 100%):

PVDF: _____

PTFE: _____

Part 4: Key Technical Details of Curtain-type Membrane

1. What is the output proportion of products with inner liner tube/central support tube among all curtain-type membranes produced by your company?

< 20%

20% - 50%

50% - 80%

> 80%

Almost all (>95%)

accurate data: _____

2. What are the main advantages of producing curtain-type membranes with inner liner tubes (multiple choices allowed)?

Enhanced Filament Strength

High-Intensity Cleaning Requirement

Extended Module Lifespan

Meeting Project Reliability Requirements

Others: _____

Part 5: Market Application Distribution

Please estimate the main application fields and their proportions of MBR membrane products produced by your company (total 100%):

Municipal Wastewater: _____

Industrial Wastewater: _____

□ *Agricultural Wastewater:* _____

□ *Decentralized Rural Sewage:* _____

□ *Others (e.g., Drinking Water, Seawater/Brackish Water Desalination, Reclaimed Water):* _____

Q4: Concern#3: It is unclear whether, once the PVDF layer is partially removed to achieve the desired exposure, the exposed substrate remains stable over time. Hydraulic forces and biological activity could enlarge these gaps, potentially leading to excessive removal of the PVDF layer and resulting in unintended substrate exposure, such as the 50% exposure observed in Figure 4, thereby affecting membrane performance.

Response: We highly appreciate your insightful concern. Your worry is entirely reasonable. Partial removal of the PVDF layer may indeed risk further spontaneous peeling of the remaining separation layer, and long-term hydraulic forces and biological activity could alter the substrate structure. To systematically address this issue, we conducted targeted stability verification experiments from two aspects, with detailed results as follows:

1. Stability verification of the remaining PVDF layer

To confirm that the partially peeled PVDF layer does not undergo excessive spontaneous peeling during long-term operation, we performed two sets of experiments. (1) Long-term operation inspection in pilot-scale system: We examined the CTDM modules from the 20-ton/day pilot-scale experiment (AAO+MBR process, continuous operation for 45 days). Photographic observations (Supplementary Fig. S18) show that there was no obvious natural peeling of the remaining PVDF separation layer after 45 days of operation, and the designed 5% support layer exposure area remained unchanged. (2) Accelerated stability tests: We prepared short membrane filaments (1 cm in length, half with PVDF layer peeled off, half intact) and subjected them to accelerated aging test: Ultrasonic oscillation test (10 min on/10 min off cycle) and NaClO immersion test for 48 hours. Photographs of the membrane filaments before and after the tests (Supplementary Fig. S19 and S20) show no significant differences in the boundary between the peeled and intact areas, and no additional PVDF layer peeling was observed.

These results collectively confirm that the remaining PVDF layer maintains good stability during long-term operation and under simulated extreme conditions, without the risk of excessive spontaneous peeling.

2. Stability verification of the support layer structure (against pore enlargement and structural damage)

We also concerned that long-term biological activity and hydraulic forces may enlarge the pores of the support layer, thereby affecting filtration performance. To verify this, we characterized the support layer of CTDM modules at three key time points (0 day, 20 days, and 45 days of operation) using scanning electron microscopy (SEM). The SEM images (Supplementary Fig. S20) show that there were no significant differences in the porous network structure of the support layer among the three time points: the pore size distribution remained uniform, and the woven skeleton structure of the support layer did not show any structural damage (e.g., fracture, deformation) or obvious pore enlargement. This indicates that the support layer itself has excellent structural stability, and long-term operation under the combined action of microbial activity and hydraulic forces will not cause changes in its structural morphology that affect filtration performance.

In summary, the above two aspects of experiments fully confirm that both the remaining PVDF layer and the exposed support layer can maintain good stability during long-term operation, which effectively addresses the concerns about the long-term stability of the CTDM system.

Action:

1. The following paragraph has been added:

Line 307-350: To validate the practical feasibility of the CTDM technology in real-world municipal wastewater scenarios, a 45-day pilot-scale experiment was conducted at the Phase IV Facility of Yuhang Wastewater Treatment Plant (Hangzhou, Zhejiang, China), using a 20-ton-per-day AAO+MBR treatment unit (Supplementary

Fig. 16). The CTDM system was assembled from 6-year-old waste MBR membranes with 5% internal substrate layer exposure, and its performance was compared with a conventional intact MBR system. Results showed that while CTDM effluent turbidity was slightly higher than that of intact MBR membranes (still meeting general application requirements), there was no significant difference in COD, $\text{NH}_4^+\text{-N}$, TP, and SS removal—attributed to pollutant removal primarily relying on microbial metabolism in the bioreactor (Supplementary Fig. 16c). Furthermore, Dynamic membranes have been demonstrated to effectively remove various types of pollutants, including sulfonamide antibiotics, heavy metals, dissolved organic matter, and sub-micron-sized polystyrene microplastics, in wastewater treatment processes³⁷⁻³⁹.

Notably, long-term operation verified two key stability aspects: the core woven skeleton of the internal substrate layer remained structurally intact (no skeleton damage, pore collapse, or structural deformation. Supplementary Fig. 17), and the 5% peeled PVDF interface (separation between the PVDF layer and internal substrate layer) showed no unintended detachment or expansion (Supplementary Fig. 18). These results confirm that under practical operational conditions, the CTDM system maintains both internal substrate layer structural stability and PVDF-internal substrate layer interface stability, while sustaining consistent pollutant removal and filtration performance. In addition, after dynamic membrane formation, similarly high effluent quality has been reported for supports with pore sizes ranging from 5 to 50 μm ⁴⁰. This supports the finding that an effective dynamic layer can still form on the internal substrate layer despite potential minor changes in its pore size or morphology.

As the CTDM technology relies on the existing MBR operational framework, it must be compatible with conventional MBR cleaning regimes (intermittent backwashing, low-frequency chemical cleaning, and mild mechanical scrubbing). To evaluate whether conventional MBR cleaning methods affect CTDM's stability, experiments were conducted targeting the internal substrate layer structure and PVDF-internal substrate layer interface: a 48-hour cyclic ultrasonic test (10 min on/10 min off) and a 48-hour NaClO immersion test. Results showed neither the substrate layer structure nor the peeled PVDF interface exhibited unintended changes (e.g.,

interface detachment, substrate layer deformation; Supplementary Figs. 19–20), indicating that conventional MBR cleaning methods have relatively minor impacts on both aspects of CTDM stability. Although our experiments demonstrate that conventional MBR cleaning methods (low-frequency chemical cleaning and mechanical scrubbing) have a relatively minor impact on PVDF layer integrity under the tested conditions, potential cumulative effects may arise from long-term, high-frequency, or high-intensity cleaning in practical applications. Additionally, while continuous PVDF peeling might occur during extended operation, results from Figs. 4a–d suggest that even with greater peeling, the CTDM system’s filtration performance remains stable, further supporting its practical applicability.

2. The following Figures have been added:

Supplementary Fig. 16 Pilot-scale comparison of effluent quality between MBR and CTDM processes. (a) Photograph of the 20-ton-per-day pilot-scale setup. (b)

Flowchart of the integrated wastewater treatment system. (c1-c6) Temporal variations in key water quality parameters (COD, $\text{NH}_4^+\text{-N}$, TN, TP, Turbidity, and pH) of the influent, MBR effluent, and CTDM-5% effluent over a 20-day operation.

Supplementary Fig. 17 Morphological evolution of exposed support layer of waste MBR membrane during long-term operation. (a-c) Photographic image and SEM images of the support layer immediately after exposure 0 days, 20 days, and 45 days of operation.

Supplementary Fig. 18 Characterization of the 5% exposed support layer interface of CTDM membrane before and after long-term operation. (a) Dimensional record of the 5% peeled area of the membrane prior to system startup. (b) Dimensional record of the 5% peeled area of the membrane after 45 days of continuous operation. (c) SEM images of the peeled interface of the membrane after 45 days of operation (focusing on the state change of the interface).

3. The following method has been added:

Supplementary Method 2

Pilot-Scale Experimental Setup and Operation

The pilot-scale system was deployed at the Phase IV Facility of Yuhang Wastewater Treatment Plant (Hangzhou, Zhejiang, China), with a designed treatment capacity of 20 t/d. The unit processed actual municipal wastewater from the plant's influent pipeline, ensuring alignment with real operational conditions. The system adopted an integrated AAO+MBR process (Supplementary Fig. 15b), comprising a regulating tank, anaerobic tank, anoxic tank, aerobic tank, and membrane tank (equipped with parallel intact MBR modules and CTDM modules). New PVDF hollow-fiber membranes (matching the original specifications of the plant's decommissioned units) as the control group.

CTDM Fabricated from 6-year-old waste MBR membranes (collected from the plant's decommissioned units). The CTDM system consisted of 5 independent

modules, each containing 36 hollow-fiber membrane filaments (80 cm in length per filament). For each filament, the PVDF separation layer was peeled off in equal proportion (via precision blade cutting and digital caliper area measurement) to achieve a uniform total support layer exposure area of 5% across all filaments and modules.

The system operated continuously under typical municipal wastewater treatment conditions:

- Hydraulic retention time: 12 h (anaerobic tank: 2 h, anoxic tank: 3 h, aerobic tank: 6 h, MBR tank: 1 h).
- Aerobic tank MLSS: 3300 ± 200 mg/L.
- Aeration intensity (aerobic tank): $2.5 \text{ m}^3/(\text{m}^2 \cdot \text{h})$.
- Filtration flux (MBR/CTDM): 15 LMH.

To systematically track the structural stability of the membrane during operation, membrane samples were collected at three key time points (0 day, 20 days, and 45 days of continuous operation) from the CTDM system. The collected samples were first photographed to record the macroscopic surface state (e.g., PVDF layer integrity, biofilm attachment), and then subjected to scanning electron microscopy (SEM) characterization to observe the microscopic structural changes of the support layer (e.g., pore size distribution, skeleton integrity) and the PVDF separation layer. This monitoring strategy ensures the timely capture of potential structural changes of the membrane during long-term operation, providing direct evidence for evaluating the stability of the partially peeled membrane.

Q5: Concern#4: In addition, the choice of mechanical or chemical cleaning methods could significantly impact operational performance by affecting how much of the PVDF layer is removed. Different cleaning regimes may lead to uneven substrate exposure, which could alter hydraulic flow, biofilm development, and overall membrane filtration efficiency.

Response: We sincerely appreciate your insightful comment. Your concern about the potential impact of cleaning methods on PVDF layer removal and substrate exposure is highly reasonable and critical for the practical application of the CTDM technology. Here, we supplement detailed explanations and experimental evidence to address this issue in a logically coherent manner:

1. Experimental verification of cleaning method stability

It is important to clarify that the mechanical and chemical cleaning methods adopted in this study are designed to mitigate membrane fouling (e.g., sludge cake accumulation and dynamic membrane blockage) during operation in activated sludge systems. To verify whether these cleaning processes would cause unintended additional PVDF layer peeling, we conducted targeted stability experiments consistent with the Q4 response: a 48-hour cyclic ultrasonic test (10 min on/10 min off, 40 kHz) and a 48-hour 15% sodium hypochlorite soaking test. The results showed no observable significant additional detachment or expansion of the pre-controlled 5% PVDF peeled interface (Supplementary Figs. S18–19), indicating that the cleaning methods did not lead to detectable excessive PVDF layer removal.

2. Compatibility with conventional MBR cleaning regimes

It is necessary to emphasize the cleaning regime of conventional MBR systems, as the CTDM technology is developed based on end-of-life (EoL) MBR membranes and relies on the existing MBR operational framework for engineering application. Conventional MBR operations typically adopt a two-stage cleaning mode: daily intermittent clean water backwashing (to remove loose sludge cake) and

low-frequency chemical cleaning (once every several months to a year, to address irreversible fouling). To ensure compatibility with existing MBR infrastructure and reduce engineering modification costs, the CTDM system must maintain consistency with this conventional cleaning regime. As verified by our supplementary experiments, the low frequency of chemical cleaning and mild intensity of backwashing have a relatively minor impact on PVDF layer integrity, alleviating concerns about uneven substrate exposure caused by cleaning.

3. Applicability of mild mechanical cleaning in pilot-scale experiments

Furthermore, during the 45-day pilot-scale experiment, we applied mild mechanical cleaning (brush scrubbing) to address membrane fouling after blockage. Experimental observations and quantitative analysis indicated that this mild mechanical cleaning did not result in observable significant additional PVDF layer peeling, further supporting the compatibility of routine MBR cleaning methods with the CTDM system.

Given the above experimental findings, we have supplemented relevant explanations in the manuscript to explicitly state the limitations of the CTDM technology. Specifically, while our experiments demonstrate that conventional MBR cleaning methods (low-frequency chemical cleaning, mild mechanical scrubbing, intermittent backwashing) have a relatively minor impact on PVDF layer integrity under the tested conditions, potential cumulative effects may arise from long-term, high-frequency, or high-intensity cleaning in practical applications. We have clearly noted this limitation in the manuscript and emphasized that in engineering practice, it is necessary to maintain consistent cleaning intensity and frequency with conventional MBR systems to avoid unintended excessive PVDF layer removal.

Action:

1. The following paragraph has been added:

Line 321-332: Notably, long-term operation verified two key stability aspects: the core woven skeleton of the internal substrate layer remained structurally intact (no

skeleton damage, pore collapse, or structural deformation. Supplementary Fig. 17), and the 5% peeled PVDF interface (separation between the PVDF layer and internal substrate layer) showed no unintended detachment or expansion (Supplementary Fig. 18). These results confirm that under practical operational conditions, the CTDM system maintains both internal substrate layer structural stability and PVDF-internal substrate layer interface stability, while sustaining consistent pollutant removal and filtration performance. In addition, after dynamic membrane formation, similarly high effluent quality has been reported for supports with pore sizes ranging from 5 to 50 μm ⁴⁰. This supports the finding that an effective dynamic layer can still form on the internal substrate layer despite potential minor changes in its pore size or morphology.

Line 333-350: As the CTDM technology relies on the existing MBR operational framework, it must be compatible with conventional MBR cleaning regimes (intermittent backwashing, low-frequency chemical cleaning, and mild mechanical scrubbing). To evaluate whether conventional MBR cleaning methods affect CTDM's stability, experiments were conducted targeting the internal substrate layer structure and PVDF-internal substrate layer interface: a 48-hour cyclic ultrasonic test (10 min on/10 min off) and a 48-hour NaClO immersion test. Results showed neither the substrate layer structure nor the peeled PVDF interface exhibited unintended changes (e.g., interface detachment, substrate layer deformation; Supplementary Figs. 19–20), indicating that conventional MBR cleaning methods have relatively minor impacts on both aspects of CTDM stability. Although our experiments demonstrate that conventional MBR cleaning methods (low-frequency chemical cleaning and mechanical scrubbing) have a relatively minor impact on PVDF layer integrity under the tested conditions, potential cumulative effects may arise from long-term, high-frequency, or high-intensity cleaning in practical applications. Additionally, while continuous PVDF peeling might occur during extended operation, results from Figs. 4a–d suggest that even with greater peeling, the CTDM system's filtration performance remains stable, further supporting its practical applicability.

2. The following Figures have been added:

Supplementary Fig. 19 Stability verification of the PVDF layer peeled interface via ultrasonic test. (a) Appearance of the membrane (with black marking line adjacent to the peeled interface) prior to ultrasonic treatment. (b) Appearance of the membrane after 48-hour cyclic ultrasonic treatment (No obvious changes of the black marking line are observed).

Supplementary Fig. 20 Stability verification of the PVDF layer peeled interface via NaClO immersion test. (a) Appearance of the membrane (with black marking line adjacent to the peeled interface) prior to immersion treatment. (b) Appearance of the membrane after 48-hour immersion treatment (No obvious changes of the black marking line are observed).

3. The following method has been added:

Supplementary Method 3

Ultrasonic and NaClO Immersion Stability Tests

Short membrane filaments (1 cm in length, half with PVDF layer peeled off, half intact) were prepared as test samples, with three parallel samples for each treatment group. Ultrasonic treatment: Samples were placed in an ultrasonic cleaner (power: 100 W, frequency: 40 kHz) for cyclic oscillation treatment, with a cycle of 10 min on/10 min off, and total treatment duration of 48 hours. Sodium hypochlorite immersion treatment: Separate samples were immersed in a 15% sodium hypochlorite solution (consistent with the chemical cleaning concentration in the main experiment) for 48 hours at room temperature ($22 \pm 3^\circ\text{C}$), with gentle stirring every 12 hours to ensure uniform contact.

Before and after each treatment, the samples were photographed with a digital camera to observe the boundary integrity between the peeled and intact areas of the PVDF layer. For further microscopic verification, selected samples were subjected to scanning electron microscopy (SEM) characterization to evaluate the structural stability of the PVDF-support layer interface.

4. Other newly added revisions:

Supplementary Method 2 and Supplementary Figs. 16–18 have been included in the Action section of Question 4.

Q6: Concern#5: It is not clear how the authors intend to address the issue that EoL membranes may contain traces of chemical cleaning agents or tightly bound contaminants, which are difficult to remove completely. Such residues could interfere with biofilm formation or potentially compromise permeate quality, but the manuscript does not explain how these risks would be mitigated.

Response: We sincerely appreciate your insightful comment. Your concern about potential chemical residues and tightly bound contaminants in end-of-life (EoL) membranes, and their impacts on biofilm formation and permeate quality, is critical for validating the CTDM technology.

First, it should be clarified that the presence of refractory contaminants (e.g., EPS and metal cation precipitates) on MBR membrane surfaces is a common challenge faced by all MBR wastewater treatment plants, and this issue has been extensively studied in the existing literature. To mitigate this problem, the mainstream industrial practice is periodic chemical cleaning, typically using citric acid soaking (to remove metal cation precipitates) and sodium hypochlorite soaking (to degrade microbial EPS and organic contaminants). In our study, all EoL membrane samples were preprocessed using this standard chemical cleaning protocol before subsequent experiments, specifically, soaking in citric acid and sodium hypochlorite solutions sequentially, to maximize the removal of conventional contaminants and minimize interference from residual impurities.

Second, we acknowledge that trace residues may still exist after pretreatment, as you correctly noted. However, the refractory contaminants and potential chemical residues are mainly trapped in the pores of the dense PVDF filtration layer, which is the layer intentionally peeled off in the CTDM technology.

Finally, our experimental characterizations further confirm the low impurity content of the support layer. SEM observations of the EoL membrane support layer (after PVDF layer peeling) clearly show minimal surface impurities (Supplementary Fig. S4, S7-12). Meanwhile, FTIR spectroscopy analysis reveals a relatively simple surface

functional group composition of the support layer, with no characteristic peaks corresponding to residual contaminants (e.g., EPS or metal complexes) detected. These results collectively indicate that the preprocessing of EoL membranes (standard chemical cleaning) combined with the peeling of the PVDF filtration layer (the main carrier of residues) has effectively mitigated the risk of residual contaminants. Thus, the support layer, as the core functional substrate of the CTDM system, will not cause significant interference to biofilm formation or compromise permeate quality.

To fully address this concern, we will revise the manuscript as follows: explicitly describe the EoL membrane cleaning process in Materials and Methods, refine SEM/FTIR discussion with detailed observations, and Supplement a brief explanation in the Discussion section to link the EoL membrane pretreatment process with the CTDM technology's core design (PVDF layer peeling).

Action:

The following paragraph has been modified:

Line 190-199: Building on this foundation, we systematically evaluated the feasibility of utilizing exposed substrate layers as CTDM substrates. The internal substrate layers feature a micron-scale structure and have significantly lower porosity, surface area, and adsorption capacity. They lack special pollutant-intercepting functional groups, thereby lacking the ability to filter or adsorb microorganisms on their own (Supplementary Fig. 6). Prior to evaluation, the EoL membranes were pretreated with citric acid and sodium hypochlorite to remove surface impurities. FTIR spectroscopy confirmed the absence of peaks of residual contaminants (e.g., EPS or metal complexes), while SEM observations verified a clean surface of the substrate layer, eliminating interference from residual pollutants for subsequent application.

Line 492-501: Prior to use, the EoL membranes underwent sequential chemical pretreatment to remove surface and pore-blocking contaminants: first, they were soaked in a 10% (v/v) citric acid solution at room temperature ($22 \pm 3^\circ\text{C}$) for 12 h to

dissolve metal cation precipitates; subsequently, they were immersed in a 15% (v/v) sodium hypochlorite solution for an additional 12 h to degrade microbial extracellular polymeric substances (EPS) and organic pollutants. During each soaking step, the solution was gently agitated every 6 h to ensure uniform contact between the membrane and the cleaning agent. After chemical pretreatment, the membranes were thoroughly rinsed with deionized water until the effluent pH reached neutrality (6.8–7.2), then dried at 40°C for 24 h prior to subsequent PVDF layer peeling and experimental assembly.

Q7: On the other hand, minor remarks should be also considered. 1. There is a typo in Figure 2e, see "partivle".

Response: We have addressed the specific error you identified, and to ensure the overall rigor of the paper, we conducted a comprehensive full-text review to identify and correct other potential inconsistencies.

Action:

1. The following Figure has been modified:

Fig. 2. Filtration mechanism and filtration performance of CTDMs. (a) Schematic diagram of the directional transformation process from KL to AAGS. (b) Photograph and fluorescence microscopy image of the membrane filament surface under live-dead fluorescence staining (live cells appear green; dead cells appear red). (c) and (d)

Trans-membrane pressure and turbidity of curtain-type DM layer when they are operating in the MBR system at a flux of 15 LMH. (e) effluent particle size of curtain-type DM at different times. (f) and (g) Microbial content and EPS content on the membrane surface, respectively. (h) Filtration resistance model of curtain-type DM.

2. The grammar of the entire text has been checked:

We have addressed the language issues by thoroughly revising the manuscript. All grammatical edits have been tracked and are highlighted in green for your review.

3. The following references have been modified:

Reference 28 has been replaced:

30. Fan, Z. W., Xiao, C. F., Liu, H. L., Huang Q. L. & Zhao J. Structure design and performance study on braid-reinforced cellulose acetate hollow fiber membranes. *J. Membrane. Sci.* **486**, 248–256 (2015).

The following References have been deleted:

3. Smith, J. C. V., Gregorio, D. O. & Talcott, R. M. The use of ultrafiltration membrane for activated sludge separation. *Presented paper at 24th Annual Purdue Industrial Waste Conference* (1969).

22. Ling, S. J. et al. Design and function of biomimetic multilayer water purification membranes. *Sci. Adv.* **3**, e1601939 (2017).

28. Raja, N. R. S. et al. Supported liquid membrane extraction of nickel using stable composite SPEEK/PVDF support impregnated with a sustainable liquid membrane. *J. Hazard. Mater.* **15**, 120895. (2019).

36. Silva, I. P. D. et al. Transfiguration of discarded PVDF ultrafiltration membranes: optimization of pyrolysis parameters for high-value char production. *Waste Manage.* **190**, 360-369 (2024).

The following References have been added:

37. Sun, Y. Q. et al. Anaerobic dynamic membrane bioreactor treating swine wastewater: Fate of sulfonamide antibiotics and heavy metals with their effect on filtration performance. *J. Hazard. Mater.* **489**, 137718 (2025).
38. Zhang, Y. L., Zhang, H. Chu, H. Q., Zhou, X. F. & Zhao, Y. Y. A Characterization of dissolved organic matter in a dynamic membrane bioreactor for wastewater treatment. *Chinese Sci. Bull.* **58**, 1717–1724 (2013).
39. Kim, B., Lee, S. W., Jung, E. M. & Lee, E. H. Biosorption of sub-micron-sized polystyrene microplastics using bacterial biofilms. *J. Hazard. Mater.* **458**, 131858 (2023).
40. Cai, D. L. et al. Effect of support material pore size on the filtration behavior of dynamic membrane bioreactor. *Bioresource Technol.* **64**, 105673 (2024).